# Nucleoporin TPR is an integral component of the TREX-2 mRNA export pathway

Vasilisa Aksenova[1], Alexandra Smith[1], Hangnoh Lee[1], Prasanna Bhat [2], Caroline Esnault[3], Shane Chen[1], James Iben[4], Ross Kaufhold[1], Ka Chun Yau [1], Carlos Echeverria[1], Beatriz Fontoura [2], Alexei Arnaoutov[1] & Mary Dasso [1]✉

Nuclear pore complexes (NPCs) are important for cellular functions beyond nucleocytoplasmic trafficking, including genome organization and gene expression. This multi-faceted nature and the slow turnover of NPC components complicates investigations of how individual nucleoporins act in these diverse processes. To address this question, we apply an Auxin-Induced Degron (AID) system to distinguish roles of basket nucleoporins NUP153, NUP50 and TPR. Acute depletion of TPR causes rapid and pronounced changes in transcriptomic profiles. These changes are dissimilar to shifts observed after loss of NUP153 or NUP50, but closely related to changes caused by depletion of mRNA export receptor NXF1 or the GANP subunit of the TRanscription-EXport-2 (TREX-2) mRNA export complex. Moreover, TPR depletion disrupts association of TREX-2 subunits (GANP, PCID2, ENY2) to NPCs and results in abnormal RNA transcription and export. Our findings demonstrate a unique and pivotal role of TPR in gene expression through TREX-2- and/or NXF1-dependent mRNA turnover.

[1] Division of Molecular and Cellular Biology, National Institute of Child Health and Human Development, National Institutes of Health, Bethesda, MD 20892, USA. [2] Department of Cell Biology, University of Texas Southwestern Medical Center, Dallas, TX 75390, USA. [3] Bioinformatics and Scientific Programming Core, National Institute of Child Health and Human Development, National Institutes of Health, Bethesda, MD 20879, USA. [4] Molecular Genomics Core, National Institute of Child Health and Human Development, National Institutes of Health, Bethesda, MD 20879, USA. ✉email: dassom@mail.nih.gov

Eukaryotic mRNAs are extensively processed, undergoing 5′-capping, splicing, and 3′-cleavage, followed by 3′-poly-adenylation. A series of evolutionarily conserved complexes are recruited to nascent mRNA transcripts co-transcriptionally and escort processed messenger ribonucleoprotein complexes (RNPs) to the nuclear export gates, called nuclear pore complexes (NPCs)[1,2]. These various steps are functionally linked; a failure to perform any of them during mRNA biogenesis will directly affect both upstream and downstream events[3]. A key player in mRNA maturation is the TRanscription and EXport 2 (TREX-2) complex, which bridges the transcription and export machineries in yeast through association with the Mediator complex and the NPC[4]. The TREX-2 complex is evolutionarily conserved between yeast and humans[5]. The human TREX-2 complex consists of GANP (germinal center-associated nuclear protein, scaffolding subunit) (Saccharomyces cerevisiae homolog = Sac3), PCID2 (Thp1), ENY2 (Sus1), DSS1 (Sem1), and CETN2/3 (Cdc31) proteins, where GANP serves as a scaffolding platform for other subunits[5]. In yeast and humans, loss of TREX-2 components leads to defects in mRNA export[6–11], similar to the phenotype observed after loss of major mRNA export receptor NXF1 (Mex67)[3].

The GANP subunit of TREX-2 localizes within the nucleus[12] and associates with a NPC structure called the nuclear basket that protrudes from the nucleoplasmic face of the NPC[7,13]. It has been proposed that GANP interacts with FG-nucleoporins at the nuclear basket to facilitate export of RNA transcripts whose expression is a part of an adaptation mechanism to rapidly changing environmental conditions[6,12]. The nuclear basket is comprised of three nucleoporins (BSK-NUPs) called NUP153, TPR, and NUP50 in vertebrates[14]. BSK-NUPs have been implicated in numerous processes beyond protein import and export, including chromatin remodeling, control of gene expression, and protein modification, as well as mRNA processing and export[15–17]. TREX-2 associates with NPCs in yeast through the NUP153 homolog, Nup1[13]. In vertebrates, however, TREX-2 association with NPCs was shown to require NUP153 and TPR, although it remains unresolved whether both NUP153 and TPR are required for TREX-2 function[7,18].

It has been difficult to analyze discrete NPC functions in the absence of vertebrate BSK-NUPs; knockout of these genes is deleterious for organisms, and their depletion by RNAi requires extended incubations, potentially allowing the emergence of secondary phenotypes from prolonged NPC disruption or defective postmitotic NPC re-assembly[19–22]. As a result, details of BSK-NUPs and TREX-2 interactions within the NPC and consequences of their individual loss for gene expression remain poorly understood.

Here we use a rapid auxin-mediated degradation system to untangle the functions of individual BSK-NUPs in both nuclear basket architecture and gene expression. We show that NUP153 and TPR bound to the NPC independently of each other and that loss of individual BSK-NUPs did not destabilize the NPC. We further found that TPR, but not NUP153 or NUP50, tethers the TREX-2 complex to the NPC. Loss of NUP50, NUP153 and TPR led to unique transcriptomic responses. Importantly, transcriptomic signatures after loss of TPR were more pronounced and were similar to changes upon the loss of either the GANP subunit of TREX-2 complex or RNA-export receptor NXF1. Moreover, similar to the case of NXF1 or GANP, loss of TPR led to retention of both upregulated and downregulated mRNA transcripts within the nucleus. Taken together, our data support a unique role of TPR in transcription regulation and mRNA export through the TREX-2 complex.

## Results

**Individual BSK-NUPs can be specifically degraded by auxin-inducible system.** To selectively track and eliminate individual BSK-NUPs, we used CRISPR/Cas9 gene editing to biallelically introduce sequences encoding an auxin-inducible degron (AID) and a NeonGreen (NG) fluorescent protein to the NUP50, NUP153, and TPR endogenous genomic loci in colorectal adenocarcinoma DLD-1 cells (Fig. 1a and Supplementary Figs. 1a, b and 2). To express the TIR1 ligase, which drives ubiquitination of AID-tagged proteins, we also used CRISPR/Cas9 to insert sequences encoding infra-red fluorescent protein (IFP) linked to TIR1 through a cleavable P2A sequence at the C-terminus of the ubiquitously expressed RCC1 protein (regulator of chromosome condensation 1) (Supplementary Fig. 1c). The resulting RCC1-IFP-TIR1 fusion protein was rapidly cleaved to yield TIR1 and an RCC1-IFP fusion protein (RCC1$^{IFP}$) that we employed as a marker for chromatin in live imaging experiments. We will call the resulting cell lines NUP50$^{AID}$, NUP153$^{AID}$, and $^{AID}$TPR in this report.

Without auxin, NUP50$^{AID}$, NUP153$^{AID}$, and $^{AID}$TPR cells were viable and did not show any overt defects. We confirmed that all AID-tagged fusion proteins had the expected nuclear rim localization (Supplementary Fig. 1d), their expression did not alter the assembly of other NPC structures (Fig. 1b), and they became undetectable within an hour of auxin addition (Supplementary Fig. 2c, g, k). To assess how prolonged nucleoporin deficiency impacts cell survival[22–24], we examined viability of NUP50$^{AID}$, NUP153$^{AID}$, and $^{AID}$TPR cells in the continuous presence of auxin (Supplementary Fig. 2d, h, l). NUP50-depleted cells continued to grow, albeit somewhat more slowly than parental DLD-1 cells (Supplementary Fig. 2m) and control NUP50$^{AID}$ cells. However, depletion of NUP153 or TPR caused growth arrest, indicating that these two nucleoporins are essential for cell growth and proliferation, while the presence of NUP50 is less critical.

**Stability of the assembled NPC upon loss of individual basket nucleoporins.** NPCs assemble as mammalian cells exit from mitosis and remain relatively stable through interphase until cells enter the next prophase[25,26]. Due to the multiple cell cycles required to achieve BSK-NUP depletion in earlier experiments utilizing RNAi, changes in NPC composition observed in those experiments reflect interdependence of nucleoporins both for recruitment during post-mitotic NPC assembly and for persistence within assembled NPCs. The rapidity of AID-mediated degradation allowed us to assess the stability of nucleoporins within assembled NPCs. We assayed localization of GLFG nucleoporin NUP98, Y-complex component NUP133, and cytoplasmic fibril component RANBP2, which reside in different domains of the NPC. All of these nucleoporins maintained their localization after depletion of BSK-NUPs (Fig. 1b, Supplementary Fig. 3a, and Supplementary Data 1), and loss of individual BSK-NUPs did not cause visible nuclear envelope (NE) deformation. Our findings thus argue that loss of BSK-NUPs from assembled NPCs does not grossly impair nuclear pore architecture or composition in other NPC domains.

**TPR does not require NUP153 for its localization during interphase.** Earlier reports indicate that TPR is dispensable for NUP153 and NUP50 localization[27,28] and that NUP50 is likewise dispensable for NUP153 and TPR localization[20,22]. However, there are conflicting reports regarding whether NUP153 depletion displaces TPR from NPCs[7,19–21,29]. To address this question, we examined BSK-NUPs by tandem mass tag (TMT) and conventional matrix-assisted laser desorption ionization time-of-flight mass spectrometry (Fig. 1c), immunostaining (Fig. 1d, Supplementary Fig. 3b, c), immunoblotting (Supplementary Fig. 3d), and live imaging of cells in which other nucleoporins

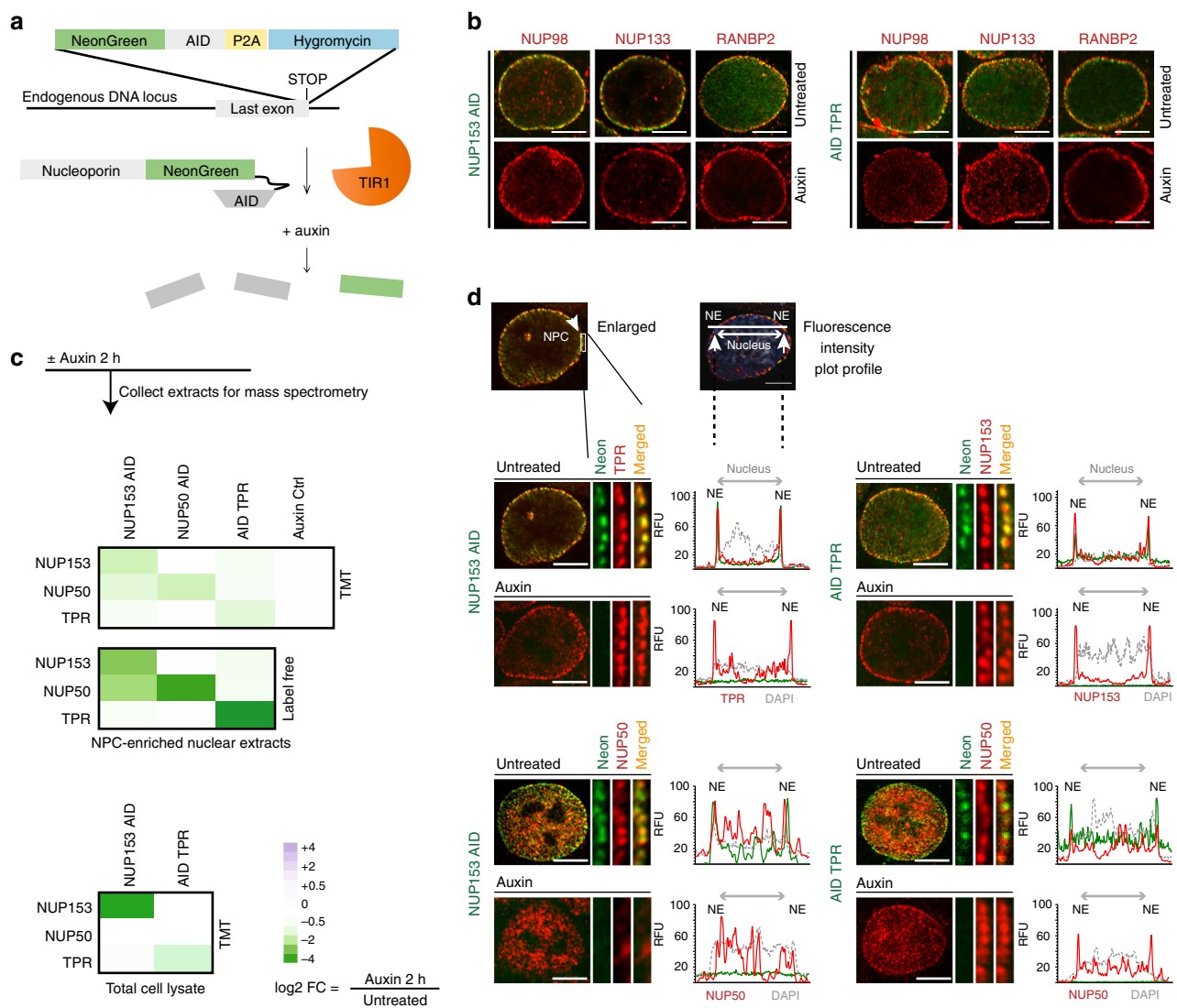

**Fig. 1 Stability of the assembled nuclear pore upon rapid loss of BSK-NUPs. a** Schematic of CRISPR/Cas9-based tagging of nucleoporin genes and subsequent degradation of the AID-fused proteins. Each AID cell line is biallelically tagged. **b** NUP98, NUP133, and RANBP2 localization is not altered upon loss of NUP153 or TPR (4 h of auxin treatment). AID-NG tagged NUP153 and TPR proteins are shown in green, antibody-stained NUPs in red. Data are representative of at least three independent experiments. **c** Heat maps of differential abundance of NUP153, NUP50, and TPR proteins in NPC-enriched nuclear extracts and total cell lysates of the indicated AID-tagged cells 2 h after auxin treatment using TMT-assisted and conventional (label-free) mass spectrometry. Green color indicates a decrease in protein abundance after auxin treatment, log2 scale. Data represent a repeat of one experiment.
**d** Localization of BSK-NUPs in the absence or presence of TPR and NUP153 (4 h of auxin treatment). Plots show line scans of relative fluorescence intensity (RFU) for BSK-NUPs (AID-NG tagged NUPs are shown in green, antibody-stained NUPs in red) and DNA across the nucleus. DNA was counterstained with DAPI (gray). Note that NUP50 no longer localizes to the nuclear envelope (NE) in the absence of NUP153. Scale bar: 5 μm. Data are representative of at least three independent experiments. AID auxin-inducible degron, TMT tandem mass tag, FC fold change.

were tagged with mCherry at their endogenous loci (Supplementary Fig. 4b–h). Under these conditions, the localization of BSK-NUPs was remarkably independent; none of them were redistributed upon the loss of others, with the exception of NUP50 dispersion into the nucleoplasm upon NUP153 loss (Fig. 1d (lower left) and Supplementary Fig. 4a, h). Importantly, TPR remained at NPCs even after several hours of NUP153 depletion, although we observed a mild reduction (~25%) of TPR abundance after NUP153 loss in unsynchronized DLD-1 cells using TMT mass spectrometry (Fig. 1c). Notably, postmitotic recruitment of TPR to NPCs was completely abolished in the absence of NUP153: these cells were characterized by the restricted nuclear growth and accumulated aggregates of TPR-mCherry in the cytoplasm (Supplementary Fig. 3e, f). Altogether, our results indicate that NUP153 is required for the recruitment

of TPR during postmitotic NPC assembly but is dispensable for anchoring of TPR that is already localized within the assembled nuclear pore (Supplementary Fig. 5a, b).

**Loss of BSK-NUPs leads to unique transcriptomic responses.** BSK-NUPs have been proposed to promote the establishment of active chromatin near the nuclear pore and thereby to help to control gene expression during development and differentiation[30]. To assess whether loss of individual BSK-NUPs alters the cellular transcriptome, we performed RNA-seq analysis to estimate changes in gene expression. We compared RNA abundance upon the loss of individual nucleoporin and found surprisingly distinct transcriptomic signatures for each constituent of BSK-NUPs (Fig. 2a, b). We detected significant changes in the

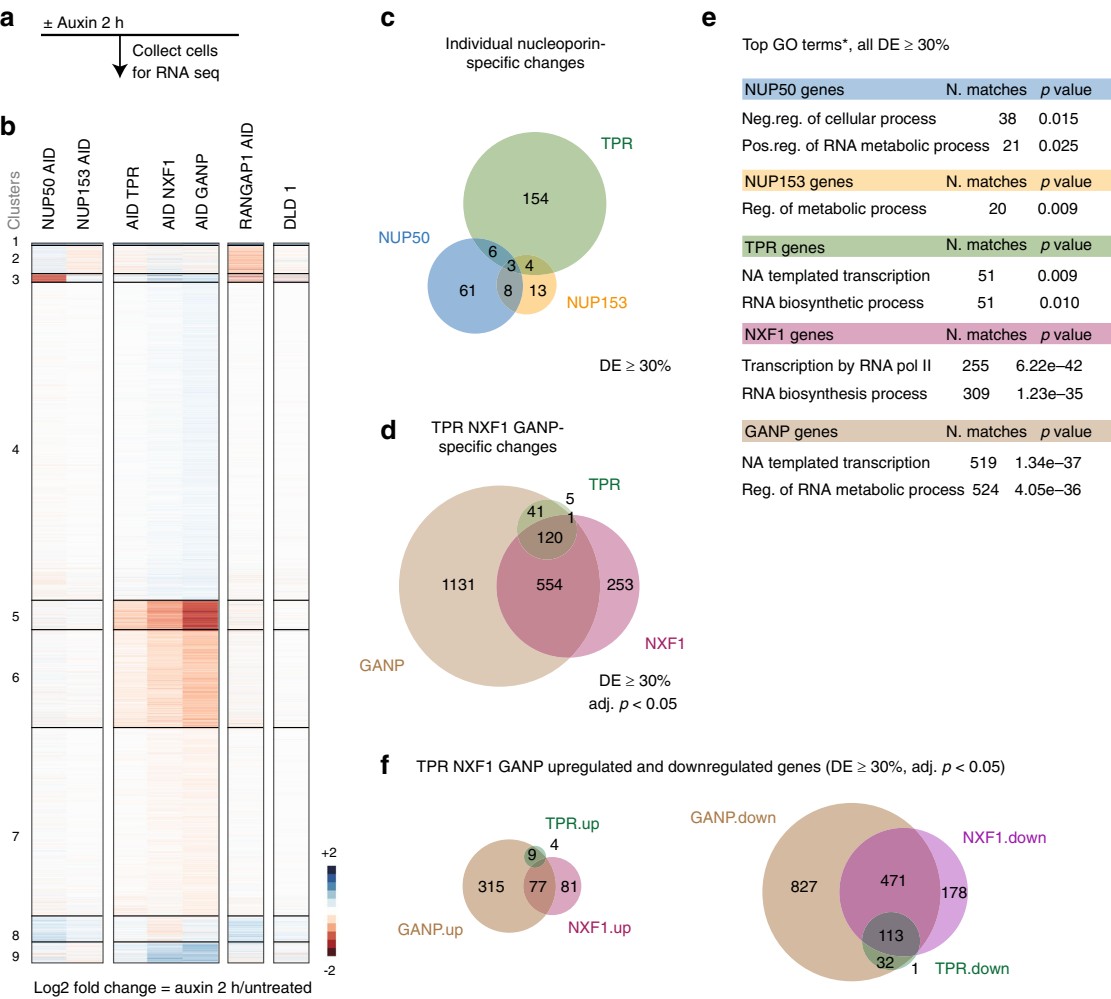

**Fig. 2 Loss of TPR leads to rapid changes in mRNA abundance. a, b** A scheme of the experiment and heatmaps of unsupervised *k*-means clustering of differentially expressed genes 2 h after auxin treatment of cells expressing corresponding AID-tagged proteins. DLD-1 parental DLD-1 cells treated with auxin. Three independent biological replicates were used to perform RNA-seq of the indicated cell lines. **c** A Venn diagram representing the number of RNAs that showed significant change (both upregulation and downregulation) upon NUP50 (blue), NUP153 (yellow), and TPR (green) loss. **d** A Venn diagram representing the overlaps among significantly regulated transcripts upon TPR, NXF1 (purple), or GANP (brown) depletion (*p* value <2.3−117, hypergeometric distribution test). **e** Top GO terms (*Biological Process, HumanMine v5.1 2018) of differentially expressed (DE) RNAs upon loss of the indicated nucleoporins (**c**), NXF1, and GANP (**d**). All DE RNAs (**c, d**) are log2FC > 30%, adj. *p* value <0.05, Wald test. **f** A Venn diagram representing the overlaps between significantly upregulated or downregulated transcripts upon TPR, NXF1, or GANP loss. Note that most of the TPR/NXF1/GANP-dependent RNAs are downregulated. Neg. negative, Pos. positive, Reg. regulation, NA Nucleic Acid, GO Gene Ontology, FC fold change.

abundance of 167 RNAs upon TPR loss (differential expression (DE) ≥ 30%, adj. *p* value <0.05), 154 of which were TPR specific, whereas loss of NUP153 or NUP50 caused DE of 28 (13 NUP153 specific) and 78 (61 NUP50 specific) genes, respectively, without extensive overlap between the DE profiles (Fig. 2b, c); these genes are grouped in visually distinct clusters (*k*-means clustering; Fig. 2b and Supplementary Data 2). Gene Ontology (GO) analysis (Fig. 2e and Supplementary Data 3) revealed that TPR loss differentially and specifically impacted RNAs involved in regulation of transcription, whereas NUP153- and NUP50-specific RNAs showed no enrichment in these categories (Fig. 2e). Interestingly, the majority of TPR-regulated RNAs were down-regulated (Supplementary Data 2).

Nuclear pores are major nuclear–cytoplasmic conduits. To determine whether blocking nuclear–cytoplasmic trafficking alters RNA abundance, we compared the impact of BSK-NUP loss to defects caused by loss of RANGAP1 (the principal GTPase activating protein for Ran-dependent nuclear transport)[31], Exportin-1 (CRM1)[32] or a major mRNA export receptor NXF1[1,2].

We constructed a RANGAP1[AID] cell line, wherein *RANGAP1* was tagged with AID and NG using a strategy similar to BSK-NUPs (Supplementary Fig. 5c–f). RANGAP1[AID] cells grew well and showed no morphological defects prior to auxin addition. RANGAP1 was undetectable within 1 h of auxin addition, and prolonged RANGAP1 depletion caused RANGAP1[AID] cells to lose their viability, as expected (Supplementary Fig. 5d). Remarkably, among 154 RNAs that changed in response to loss of RANGAP1, 111 were specific to RANGAP1[AID] cells (when compared to TPR-, NUP50-, or NUP153-dependent changes), and the majority of BSK-NUP-regulated RNAs did not overlap with the changes observed in RANGAP1-depleted cells (Supplementary Fig. 5g). This data indicates that the observed DE of BSK-NUP-dependent RNAs are not due to the overall deficit in Ran-dependent nuclear trafficking.

Further, we analyzed RNA transcripts that were affected upon inhibition of nuclear export receptor CRM1 with Leptomycin B[33]. CRM1 binds NES-containing cargos and participates in export of small nuclear RNAs, ribosomal RNAs, and a subset of mRNAs;

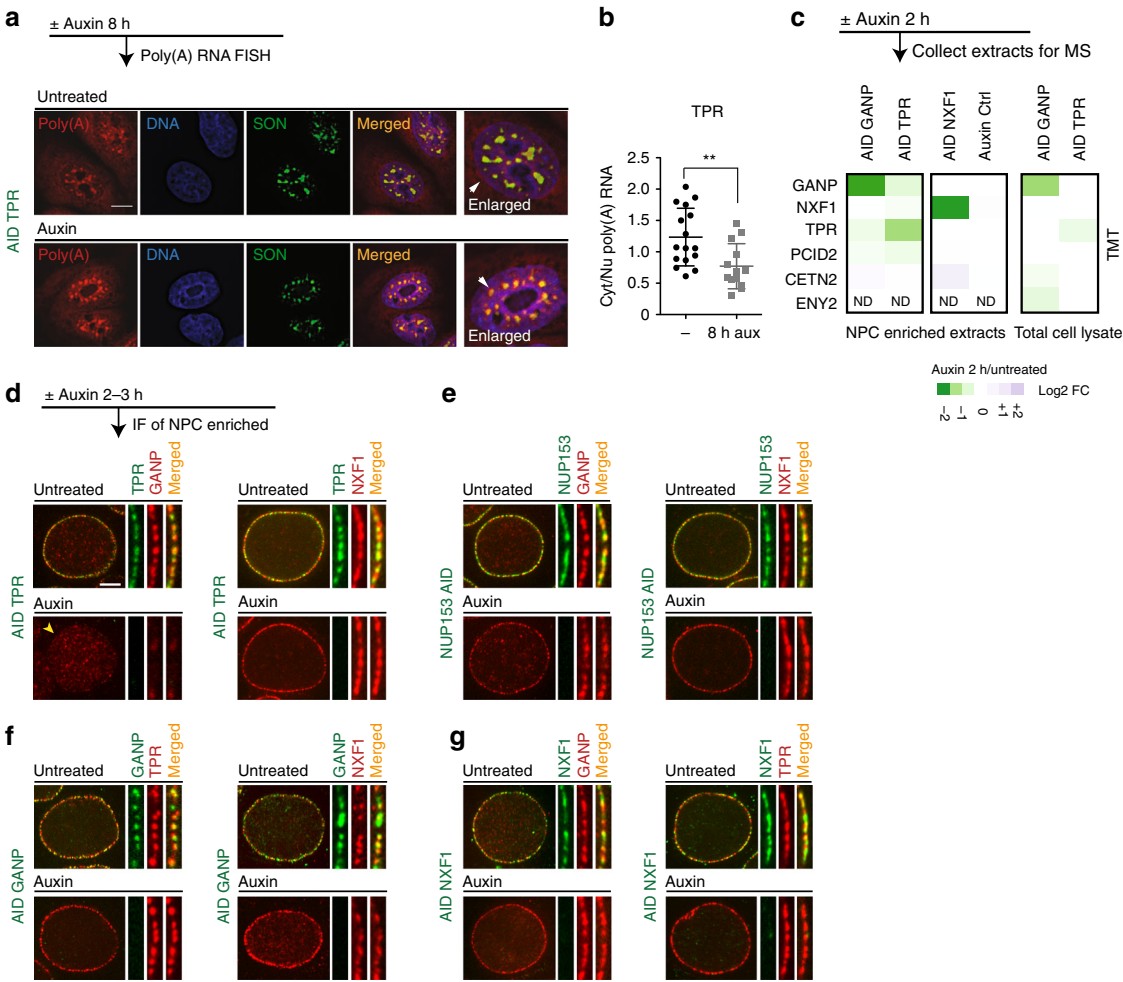

**Fig. 3 TPR is required for both GANP localization and efficient export of poly(A) RNA. a** The analysis of cytoplasmic–nuclear distribution of poly(A) RNA in ^AID^TPR cells using oligo(dT)-Quasar 670 probe 8 h after TPR degradation. Note that Poly(A) RNA accumulates in the nuclear speckles. Scale bar: 10 μm. **b** Quantification of cytoplasmic-to-nuclear (Cyt/Nu) Poly(A) RNA distribution. Biological triplicates from two independent experiments were used (**p value 0.0062, n = 16 untreated and n = 13 auxin-treated cells, unpaired two-tailed Student's t test). Data are presented as mean values; error bars are SD. **c** A heat map of differential abundance of the indicated proteins in either NPC-enriched fractions or total lysates upon loss of GANP, TPR, or NXF1, analyzed by TMT-assisted mass spectrometry. Note that the loss of TPR results in diminished abundance of TREX-2 complex subunits (GANP and PCID2) to the NPC but does not affect their total protein amount. Data represent a repeat of one experiment. **d–g** Targeting of GANP and NXF1 to the NPC. **d**, **e** GANP localization at the NE depends on TPR but not on NUP153 or NUP50, while NXF1 localization is not affected upon loss of either BSK-NUPs (see also Supplementary Fig. 7a). **f**, **g** GANP and NXF1 localization at the NE are independent of each other. Scale bar: 5 μm. All BSK-NUPs and NXF1 AID-tagged cell lines were treated with auxin for 2 h; GANP AID-tagged cell line was treated with auxin for 3 h. Data are representative of at least three independent experiments. MS mass spectrometry, IF immunofluorescence, FC fold change, NE nuclear envelope.

it is also involved in regulation of gene transcription[32,34]. Among CRM1s significantly upregulated (DE > 30%, 317 genes) and downregulated (DE > 30%, 642 genes) transcripts, only 6 upregulated and 42 downregulated transcripts had an overlap with TPR-specific RNAs (Supplementary Fig. 5h), suggesting that majority of TPR-dependent RNAs do not engage CRM1 pathway. Finally, we tested whether loss of TPR affects the dynamics of nuclear–cytoplasmic transport of a model NLS/NES-containing substrate and found that the rates of both import and export were not significantly altered in ^AID^TPR cells after auxin addition (Supplementary Fig. 6a, b).

Although TPR-dependent RNAs did not appear to be impacted by disruption of RanGTP- or CRM1-dependent export pathways, we found that long-term (8 h) loss of TPR led to a measurable retention of bulk poly(A) mRNA within the nucleus, with its specific accumulation in nuclear speckles (Fig. 3a, b). Importantly, this phenotype was not observed after depletion of NUP153, NUP50, or RANGAP1 (Supplementary Fig. 6c, d). Notably, bulk

poly(A) mRNA retention was also reported after depletion of mRNA export factor NXF1[7]. To test whether TPR-specific transcripts are altered upon NXF1 loss, we constructed ^AID^NXF1 cell line (Supplementary Fig. 6e–h). AID-NXF1 degraded within an hour of auxin addition and prolonged incubation of ^AID^NXF1 cells with auxin resulted in a rapid arrest of cell growth and loss of cell viability (Supplementary Fig. 6f). Two hours of NXF1 loss led to substantial changes in RNA abundance for 927 transcripts (Fig. 2d, DE > 30%, adj. p < 0.05). Overall NXF1-dependent transcripts overlapped with majority (121 out of 167) of TPR-dependent transcripts, strongly indicating that both proteins participate in the same pathway.

**TPR tethers TREX-2 complex to the NPC.** The functional overlap of TPR and NXF1 suggested direct TPR involvement in RNA processing and trafficking. We hypothesized that one or several RNA processing or export factors might be recruited to

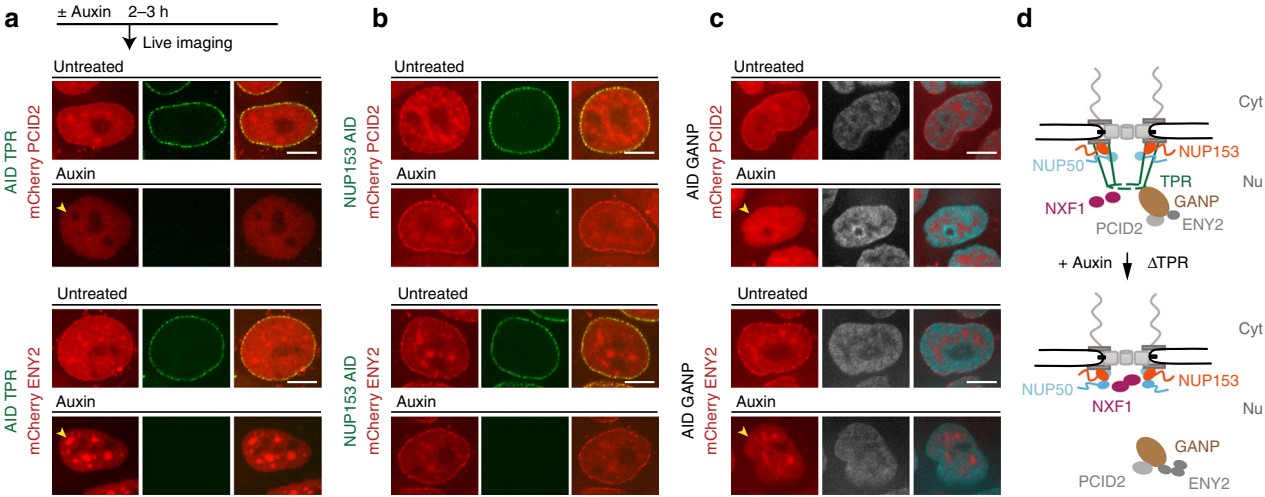

**Fig. 4 TPR is required for localization of TREX-2 complex at the NPC. a–c** Both TPR and GANP are required for PCID2 and ENY2 localization at the NE. Localization of mCherry-PCID2 and mCherry-ENY2 in cells after acute loss of TPR (**a**), NUP153 (**b**), or GANP (**c**). (see also Supplementary Fig. 7b, c). Note the changes in NPC localization of PCID2 and ENY2 (yellow arrowheads). Scale bar is 7 μm. All BSK-NUPs and NXF1 AID-tagged cell lines were treated with auxin for 2 h; GANP AID-tagged cell line was treated with auxin for 3 h. Data are representative of three independent experiments. **d** A schematic representation of the TREX-2–NPC interaction.

the NPC in TPR-dependent manner in order to perform their function. To address this, we processed NPC-enriched protein extracts by mass spectrometry and found a reduction of GANP[35] and PCID2 subunits of TREX-2 complex in TPR-depleted cells but not in NUP153- or NUP50-depleted cells (Fig. 3c and Supplementary Data 1). Notably, depletion of GANP was reported to cause poly(A) RNA accumulation in nuclear speckles[7,12,36], reminiscent of our observations in TPR-depleted cells.

To further examine the relationship between TREX-2 and TPR, we engineered an [AID]GANP cell line (Supplementary Fig. 6i–l). AID-tagged GANP degraded within 3 h of auxin addition, and prolonged incubation with auxin resulted in a rapid arrest of cell growth and loss of cell viability, similar to NXF1- and TPR-depleted cell lines (Supplementary Fig. 2l and Supplementary Fig. 6f). We compared the impact of TREX-2 or NXF1 depletion to the loss of BSK-NUPs, analyzing both protein localization and changes in the transcriptome. Immunofluorescent microscopy showed that GANP, which is normally localized to NPC, is dispersed after TPR loss but not after NUP50 or NUP153 depletion (Fig. 3d, e and Supplementary Fig. 7a), confirming the mass spectrometric results (Fig. 3c). GANP and NXF1 were each retained at the NPC in the absence of the other, and TPR localization was not dependent on either protein (Fig. 3f, g). Further analysis showed that similar to GANP, other two TREX-2 complex subunit (PCID2 and ENY2) were mislocalized from the NPC into the nuclear interior after loss of TPR and GANP (Fig. 4a, c, d) but not after loss of NUP153, NUP50 or NXF1 (Fig. 3b and Supplementary Fig. 7b–d). These data support earlier observations that GANP is required for the recruitment of other TREX-2 complex subunits[5] and highlight the requirement of TPR for TREX-2 localization at the NPC.

**TPR has a unique function in gene expression regulation of TREX-2-dependent RNAs.** Transcriptomic analysis showed that TPR-dependent RNAs overlapped almost entirely with GANP-dependent transcripts, which in turn overlapped extensively with NXF1-dependent RNAs (Fig. 2d, p value = 2.3e−117) with similar patterns of upregulation and downregulation (Fig. 2b, f). GANP-, NXF1-, and TPR-dependent RNAs also share similar GO terms (Fig. 2e and Supplementary Data 3). In combination with the largely dissimilar transcriptomic patterns observed after

NUP153 or NUP50 depletion, these data suggest that TPR has a unique function within the NPC basket to anchor GANP, PCID2, and ENY2 subunits of TREX-2 complex and facilitate export of TREX-2-dependent RNAs.

As expected, the majority of GANP- and NXF1-dependent transcripts overlap (Fig. 2d) and share similar GO terms (Supplementary Fig. 7e). Nevertheless, we noticed a subset of transcripts that were affected only after GANP loss. This group of transcripts code proteins that are involved in chromosome organization, DNA packing, and nucleosome assembly, and most of them encode histones, suggesting that GANP may have a special function in processing of these mRNAs. Individual GANP-specific RNAs were confirmed by fluorescence in situ hybridization (FISH; Supplementary Fig. 7f and Supplementary Fig. 11c), RNA-seq, and quantitative polymerase chain reaction with reverse transcription (qRT-PCR) experiments (Supplementary Fig. 7g, h).

**TPR does not affect NXF1 dynamics on the NE.** Because TPR-, TREX-2-, and NXF1-dependent transcripts overlap, we examined whether TPR is also responsible for tethering NXF1 to the NPC but did not find evidence to support this idea (Fig. 3d). We further tested whether NXF1 dynamics at the NE might be altered in the absence of TPR without detectable impact on its steady-state NPC-bound level. To address this question, we endogenously tagged NXF1 with mCherry fluorescent protein (Supplementary Fig. 8a) and measured the dynamics of mCherry-tagged NXF1 at the NE by fluorescence loss in photobleaching (FLIP) experiments. The dynamics of exchange between NPC-bound and nucleoplasmic NXF1 were similar (Supplementary Fig. 8b, c) regardless of whether TPR was present or not, strongly indicating that NXF1 localization and dynamics are independent of TPR.

**Loss of TPR leads to changes in synthesis of RNA transcripts.** To identify the primary reasons underlying changes in RNA-seq profiles upon TPR loss, we treated [AID]TPR cells with RNA-polymerase inhibitor Actinomycin D (ActD), followed by addition of auxin. ActD treatment abolished DE of most TPR-regulated RNAs in the presence of auxin, and we confirmed this observation for a subset of representative transcripts using qRT-PCR and FISH (Supplementary Fig. 9a–c and Supplementary

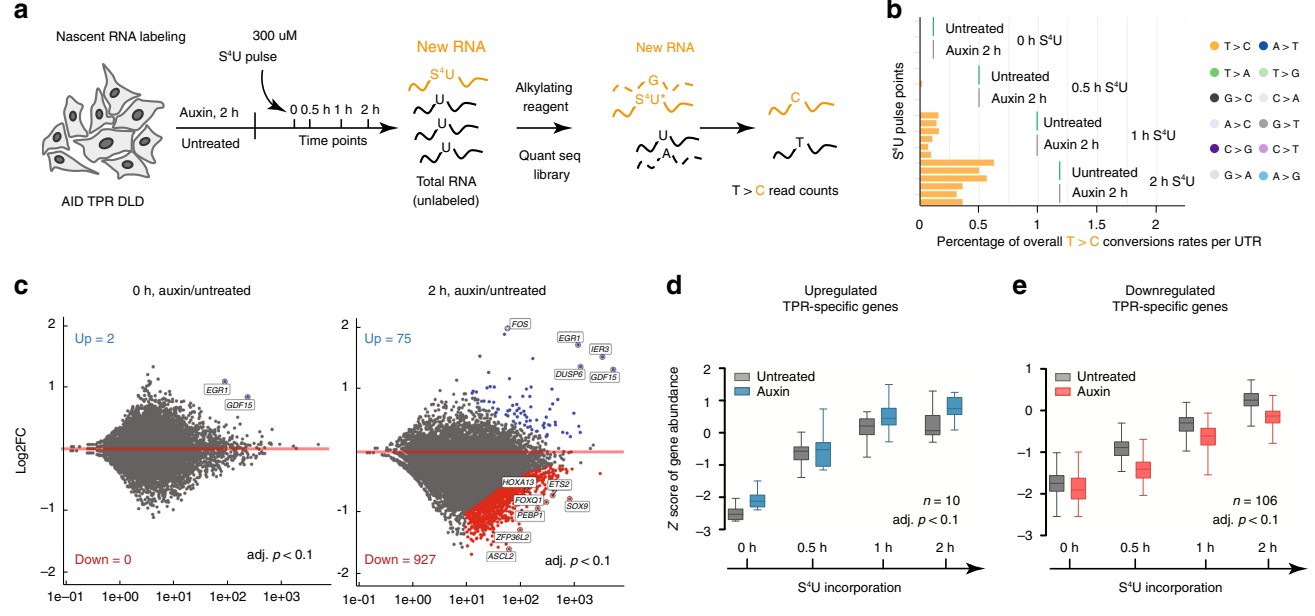

**Fig. 5 TPR regulates transcription. a** A workflow of the SLAM-seq experiment. Untreated and 2 h auxin-treated DLD-1 $^{AID}$TPR cells were labeled with S$^4$U for 0, 0.5, 1, or 2 h. 3′UTRs of purified and alkylated RNA was sequenced and T > C reads counts (marked with yellow) were analyzed by the DESeq2 (Supplementary Data 5). **b** Bar plots of the overall T > C conversion rates per UTR analyzed after Quant-Seq. Each bar plot represents an independent repeat of Quant-Seq experiment covering 19,495 genes. Note that conversion events of other nucleotides were not detectable per UTR (detailed information is provided in Source data file). **c** M-A plots representing changes in the abundance of newly synthesized transcripts after TPR loss. All significantly upregulated transcripts are marked in blue and downregulated transcripts in red (FDR-adjusted *p* value <0.1, Wald test). Top upregulated and downregulated genes are indicated. **d**, **e** Dynamics of gene transcription was analyzed for 10 upregulated (**d**) and 106 downregulated (**e**) TPR-dependent transcripts after 0, 0.5, 1, and 2 h after S$^4$U pulse labeling. Three independent replicates were used to perform SLAM-seq. Data are represented as mean values of normalized log2 T > C counts that were centered to the mean and scaled to the standard deviation to obtain *Z*-score for each gene. Details of *Z*-score quantification are described in the "Methods" section. UTR untranslated region.

Data 2). Our findings imply that the majority of changes observed in our RNA-seq data resulted from modulations of transcriptional activity. Surprisingly, only a fraction of genes whose expression changed upon TPR loss (16 out of 167) demonstrated appreciable changes in Ser5P Pol II occupancy (*p* value < 0.05) within the promoters or gene bodies of correspondent genes, as assessed using a chromatin immunoprecipitation (ChIP) assay (Supplementary Fig. 9d, e and Supplementary Data 4). However, when we analyzed upregulated and downregulated TPR genes separately, we found that approximately half of the genes that were upregulated upon TPR loss (9 out of 20) demonstrated increased Ser5P Pol II occupancy, while only few of the downregulated genes (7 out of 140) changed their Ser5P Pol II status, suggesting that TPR-downregulated genes do not change their promoter activity[37].

To further test the hypothesis that the changes in TPR-dependent RNA abundancy are due to changes in de novo synthesized transcripts, we followed the synthesis of newly transcribed RNAs using a thiol (SH)-linked alkylation for the metabolic sequencing (SLAM-seq) protocol: we supplemented cells with 4-thiouridine (S$^4$U) uridine analog for 30 min, 1, and 2 h and then analyzed 3′-untranslated regions (3′-UTRs) of newly labeled transcripts by Quant-seq (Fig. 5a). Total RNAs were processed to allow conversion of S$^4$U to cytosine, sequenced, and the T > C reads were used to identify genes that were upregulated and downregulated in response to TPR loss (Fig. 5b). Overall, we detected 1002 transcripts (adj. *p* < 0.1) that changed their expression after depletion of TPR. In agreement with our RNA-seq data, the majority of these transcripts were downregulated (2 h of labeling with S$^4$U, total of 4 h auxin addition: Fig. 5c and Supplementary Data 5). Next, we compared top upregulated and

downregulated genes from RNA-seq and SLAM-seq data sets. This analysis indicated that the changes of RNA abundance of most of TPR-dependent upregulated and downregulated genes (116 genes) resulted from increased or decreased rates of RNA synthesis, respectively (Fig. 5d, e).

**Loss of TPR leads to nuclear retention of TPR-GANP-dependent RNA transcripts.** We examined the fates of individual top TPR-regulated RNAs in more detail. Both SLAM-seq and qRT-PCR analyses confirmed that *c-fos* mRNA was upregulated in response to TPR depletion (Fig. 6b, c). Coincidentally, Ser5P Pol II occupancy was also increased (Fig. 6a, middle panel). Analysis of transcription dynamics over a 2-h time course of S$^4$U labeling indicated that the number of *c-fos* transcripts continued to rise over the course of the experiment (Fig. 6b). In addition, *c-fos* transcripts were specifically upregulated upon loss of TPR but not after loss of either NUP50 or NUP153 (Fig. 6c). To localize *c-fos* transcripts within cells, we performed in situ analysis using a RNAScope-based protocol that allows visualization of single RNA molecules[38]. Acute depletion of TPR led to obvious increase in the number of *c-fos* foci (Fig. 6g). Interestingly, those foci were mostly detected inside cell nuclei, and the ratio of cytoplasmic-to-nuclear signal substantially dropped (0.39 ± 0.09, in comparison to the ratio found in untreated cells: 1.4 ± 0.17, *p* value = 0.0022, Fig. 6h). Altogether, these data indicate that there is increased transcription of the *C-FOS* gene in TPR-depleted cells; at the same time, these newly synthesized *c-fos* transcripts were not efficiently exported from the nucleus (Fig. 6g, h), thus indicating that TPR controls both transcription and export of *c-fos* RNA. Importantly, two other top upregulated TPR-dependent

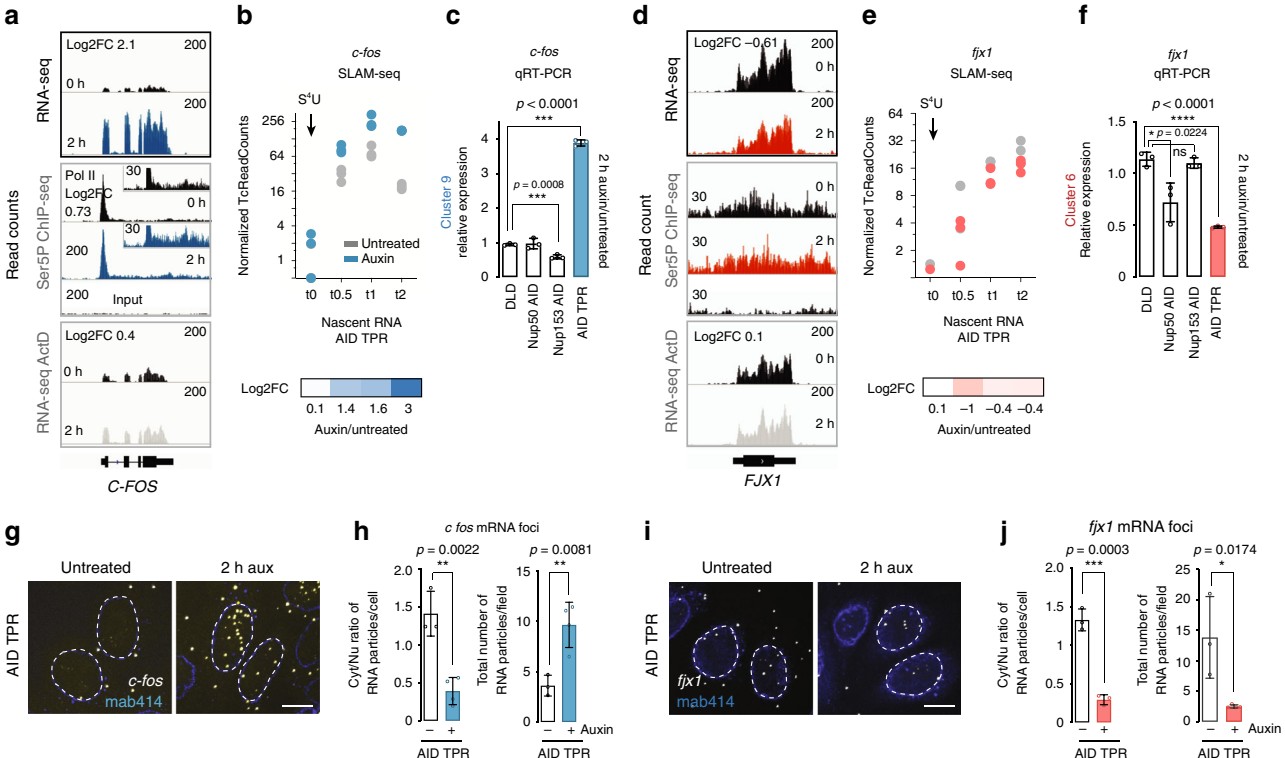

**Fig. 6 TPR controls both transcription and RNA export. a**, **e** IGV snapshots of RNA-seq, Ser5P Pol II ChIP-seq, and ActD RNA-seq results for two representative transcripts *c-fos* (blue) and *fjx1* (red). Log2FC is shown separately for each experiment. We performed three and two independent biological replicates for RNA-seq and ChIP-seq experiments, respectively. **b**, **e** Top: Normalized T > C count plots for *c-fos* and *fjx1* transcripts. Three independent replicates are shown on the plot. Bottom: Log2FC of T > C counts of auxin treated vs. untreated sample at each time point from three independent biological replicates (Supplementary Data 5). **c**, **f** qRT-PCR analysis of *c-fos* and *fjx1* RNA abundance 2 h after TPR, NUP50, or NUP153 loss. Graphs show mean values of three technical replicates of one experiment, error bars are SD. Asterisks indicate *p* value *<0.1, ***<0.001, ****<0.0001 (unpaired two-tailed Student's *t* test), ns non-significant (*p* > 0.05). Exact *p* values are indicated on the graphs and in Source data file. **g**, **i** RNA-FISH analysis of localization of *c-fos* and *fjx1* transcripts after 2 h of TPR loss. Note that both *c-fos* and *fjx1* transcripts were retained within the nucleus. Data are representative of two independent experiments. **h**, **j** Quantification of cytoplasmic to nucleus (Cyt/Nu) ratio and total number of mRNA foci after TPR loss. Data are presented as mean values, error bars are SD. Asterisks indicate *p* value *<0.1, **<0.01, ***<0.001 (unpaired two-tailed Student's *t* test). Three independent fields from two independent experiments were used for *c-fos* (untreated cells, *n* = 51; auxin-treated cells, *n* = 69 for AID-TPR cell line) and *fjx1* (untreated cells, *n* = 30, auxin-treated cells, *n* = 50) mRNA foci quantification. Exact *p* values are indicated on the graphs and in Source data file. FC fold change.

transcripts, *gdf15* and *egr1*, behaved in a similar manner (Supplementary Fig. 10a–g).

We similarly analyzed transcripts whose abundance decreased after depletion of TPR. We verified reduction in *fjx1* mRNA by qRT-PCR (Fig. 6f); this change was specific to TPR, as we did not detect *fjx1* changes after depletion of either NUP153 or NUP50 (Fig. 6f). SLAM-seq analysis demonstrated that *fjx1* synthesis was rapidly decreased after TPR depletion (Fig. 6e). However, unlike the case of TPR-dependent upregulated genes we found little or no TPR-dependent changes in Ser5P Pol II distribution along the *fjx1* locus (Fig. 6d, middle panel), suggesting that the changes in *fjx1* mRNA abundance cannot be linked to changes in Ser5P Pol II occupancy. However, because we could not detect an obvious peak of Ser5P Pol II at the *fjx1* promoter and the overall Ser5P Pol II occupancy at this locus was extremely low, it is possible that RNA transcription of this locus does not rely on Ser5 phosphorylation. RNA FISH experiments confirmed reduction of the overall number of *fjx1* transcripts after TPR depletion (Fig. 6i). Importantly, remaining *fjx1* mRNA foci resided almost exclusively within the nucleus, as the ratio of cytoplasmic-to-nuclear signal significantly diminished (1.33 ± 0.14 in control cells vs. 0.29 ± 0.06 in auxin-treated cells, Fig. 6j). Similar patterns of RNA behavior were observed for two other strongly downregulated transcripts, *ascl2* and *hoxa13* (Supplementary Fig. 10h–n). Taken together, our

data suggest that RNAs whose abundance is decreased after loss of TPR require this nucleoporin for efficient transcription-coupled processing and export.

Notably, intracellular distribution patterns of all tested TPR-dependent mRNAs (*c-fos*, *gdf15*, *hoxa13*, and *fjx1*) closely mimicked their patterns when either GANP or NXF1 were depleted (Supplementary Fig. 11a), strongly indicating that TPR, TREX-2, and NXF1 operate in the same RNA pathway. We thus observed a consistent pattern in which majority of TPR-specific genes undergo rapid transcriptional changes and both downregulated and upregulated mRNAs failed to be translocated to the cytoplasm in the absence of TPR, GANP, or NXF1.

## Discussion

In this study, rapid depletion of individual AID-tagged BSK-NUPs allowed us to demonstrate discrete functions of these nucleoporins with a temporal resolution that was previously not attainable. In particular, we found that BSK-NUPs could be depleted from the pre-assembled nuclear pores independently of each other, except NUP50, whose localization depends on NUP153. Because loss of BSK-NUPs did not destabilize the overall nuclear pore composition in the assembled nuclear pores, it was possible to analyze the specific functions of individual BSK-NUPs.

Notably, our data explain earlier contradictory findings regarding TPR localization at the NPC after depletion of NUP153[2,9–11,19]: TPR remained normally localized at the NPC despite NUP153 depletion from the assembled pores (Fig. 1d). However, loss of NUP153 led to failure of TPR to be recruited back to the NPC during NPC reassembly in postmitotic cells, causing formation of cytosolic aggregates of TPR (Supplementary Fig. 3f). We speculate that the inability of TPR to associate with NUP153-depleted postmitotic NPCs might reflect failed nuclear import of TPR. Notably, we detected a slight decrease in TPR abundance after NUP153 loss in mass spectrometric samples prepared from asynchronous cells (Fig. 1c). It will be of interest to determine whether this change solely reflect a failure of postmitotic recruitment of TPR to the NPC or whether this change also results from some requirement for NUP153 in recruitment of TPR to new NPCs that are formed during expanding nuclear growth in interphase.

By reducing the time needed for depletion of nucleoporins, we identified unique transcriptomic changes associated with NUP50, NUP153, or TPR loss. Although transcriptomic profiles of NUP153-depleted mouse embryonic stem cells[21] and Drosophila SL-2 cells[39] have been reported, analysis of RNA abundance after acute loss of NUP153, NUP50, or TPR has not been documented before. Importantly, such transcriptomic responses would be unlikely to arise through altered postmitotic NPC assembly defects or prolonged putative disruption of nuclear–cytoplasmic transport—two confounding problems associated with RNAi-based depletion of nucleoporins. Interestingly, RNA-seq profiles that we obtained after acute loss of individual BSK-NUPs differ not only from each other but also from transcriptomic changes caused by defects in protein nucleocytoplasmic protein transport, resulted from loss of RANGAP1 or CRM1 function (Supplementary Fig. 5g, h).

Among the BSK-NUPs, TPR had a dominant impact on RNA abundance. Moreover, TPR-specific changes closely mimicked those that resulted from loss of either NXF1 or GANP (Fig. 2 and Supplementary Fig. 11a), indicating a close alignment of TPR function to TREX-2 and NXF1 activities in RNA expression, processing, and export. Notably, TPR-dependent changes in RNA abundance within 2 h of TPR depletion (Fig. 2b) appear to result largely from altered rates of RNA synthesis (Fig. 5). Experiments involving ActD further support this interpretation, as most observed changes in TPR-regulated RNAs were abolished in the presence of this transcription inhibitor (Supplementary Fig. 9a). However, only about 2% of all changes in RNA-seq profiles were accompanied by Ser5P Pol II recruitment to corresponding genes, indicating a role of TPR in subsequent transcription and/or transcription-coupled processing. It is conceivable that loss of TPR results in stalling of transcription complexes with unprocessed RNAs at corresponding genes or in specific nuclear bodies, similar to the cases when RNA processing factors, such as TREX-2, are depleted (reviewed in refs. [40,41]), and this will be an interesting subject for future investigation.

Consistent with its close functional alignment to TREX-2, TPR was the only nucleoporin within the basket required for NPC recruitment of GANP and other subunits of the TREX-2 complex. These observations further demonstrate an exclusive function of TPR in coupling transcription and export that is not shared with NUP153 or NUP50. We speculate that the requirement for NUP153 in tethering TPR during postmitotic NPC re-assembly may at least partially explain previous observations arguing for an essential role of NUP153 in TREX-2-dependent RNA export[7]. Intriguingly, TPR depletion altered only TREX-2 but not NXF1 localization at the NE (Fig. 3d). This raises interesting questions regarding the functional relationship of TPR and NXF1 at NPCs. Notably, recent reports suggest that NXF1-binding sites within

the NPC are not closely adjacent to TPR and that NXF1 occupies sites on the cytoplasmic side of the NPC[42]. Consistent with this report, our findings indicate that NXF1 association with NPCs is independent of TPR, and we speculate that loss of TPR pheno-copies loss of NXF1 because NXF1 operates upstream of both TPR and GANP at early steps of mRNA processing[1].

TREX-2 coordinates transcription, 3′-end processing, and mRNA export, the processes that are tightly coupled together[40], and emerging evidence suggest that NXF1 and TREX-2 regulate transcription and processing both directly and indirectly through mechanisms that contain regulatory feedback loops. Mammalian GANP/NXF1-dependent RNAs are highly enriched in transcripts that encode components of the gene expression machinery[6], and loss of the yeast GANP homolog (Sac3) similarly impacts the expression of RNAs that fall within GO terms of "transcription" and "mRNA processing"[40]. We found that TPR loss specifically impacted a group of mRNAs similar to the mammalian GANP/NXF1-dependent RNAs (Fig. 2 and Supplementary Data 3). The similarity in patterns for transcriptomic changes after loss of NXF1, TREX-2, and TPR would be consistent with their incorporation with the same feedback regulatory circuits. Importantly, despite the extensive overlap between TPR-, GANP-, and NXF1-dependent transcripts, a subset of mRNAs responded only to loss of GANP or NXF1 (Fig. 2 and Supplementary Fig. 7f). In particular, loss of GANP impacted RNAs that encode proteins involved in chromatin organization and nucleosome assembly, particularly histone genes. These RNAs are neither TPR nor NXF1 dependent, suggesting the role of GANP in controlling these mRNAs may be independent of the regulatory circuits for RNAs controlled by all three proteins.

Impacted RNAs that we examined also showed TPR dependence for RNA export. This finding is consistent with earlier observations that TPR regulates mRNA export of transcripts with retained introns in yeast and mammals[36,43]. Notably, RNAi-mediated knockdown of TPR in mammalian cells facilitates export of retroviral cis-acting constitutive transport element-containing RNAs to the cytoplasm[18,36]. In yeast, Mlp1, a TPR homolog, is required for nuclear retention of intron-containing mRNAs and pre-mRNAs leak into the cytoplasm in the absence of Mlp1[43]. Intriguingly, it has recently been reported that incorporation of additional introns into a model RNA substrate bypasses the export requirement for TPR[44]. However, we find that prolonged TPR resulted in dramatic poly(A) RNA retention within the nucleus, associated with the nuclear speckles (Fig. 3). Moreover, the most upregulated (e.g., c-fos and gdf15) and downregulated (hoxa13 and fjx1) transcripts were all retained within the nucleus after acute loss of TPR. These transcripts contain various number of introns (0–3), indicating that the mere presence of intron(s) cannot be the sole determinant that allows avoidance of TREX-2/TPR pathway. Importantly, most TPR-dependent transcripts were also retained within the nucleus in GANP- or NXF1-depleted cells, again indicating an epistatic relationship between NXF1, TREX-2, and TPR.

Overall, we propose the following working model based on our results: (a) NXF1 binds all TPR–GANP dependent transcripts at early steps of mRNA processing, (b) GANP is then recruited to RNPs that are processed by NXF1[12], and (c) TPR works as an accessory protein for TREX-2-bound RNPs. Given the singular importance of TPR in this context, it will be important to investigate in future experiments whether it acts as a landing platform for TREX-2-bound mRNA at the NPC or whether TPR plays more "active" role in searching TREX-2 particles during transcription, ensure their functionality, and then deliver them to the NPC for efficient export. In this light, it will also be important to learn whether basket NUPs' fast dynamics[45] at the NPC contributes to its involvement in RNA transcription/processing.

AID-mediated degradation will be pivotal in future analysis of this model and of many other transport and RNA-processing pathways. It will not only be instrumental for discovering architectural composition of extremely complex molecular machines but also in determining how the NPC works as a hub for integration of these pathways to each other.

## Methods

**Cell culture.** The human colorectal cancer cell line DLD-1 was cultured in Dulbecco's modified Eagle's medium (DMEM; Life Technologies) supplemented with heat-inactivated 10% fetal bovine serum (FBS; Atlanta Biologicals), antibiotics (100 IU/ml penicillin and 100 μg/ml streptomycin), and 2 mM GlutaMAX (Life Technologies) in 5% $CO_2$ atmosphere at 37 °C. Cells for gene targeting were transfected with 500 ng of donor and gRNA plasmids in ratio 1:1 using ViaFect (Promega) transfection reagent according to the manufacturer's instruction (see Supplementary Methods for details).

**Gene targeting.** RCC1 (NC_000001.11) locus was chosen to knock-in TIR1. Using CRISPR/Cas9-mediated recombination, RCC1 was targeted with a construct that contains an IFP (IFP2.0 Addgene #54785[46] or miRF670, Addgene #79987[47]), 9Myc-TIR1 (Addgene #47328[48]), and a blasticidin resistance gene (*bsr*, pQCXIB #631516, Clontech) sequences. iRFP670/IFP2.0, TIR1, and *bsr* were separated from each other by self-cleavage peptide P2A. Because the presence of TIR1 integration can potentially reduce the protein levels of AID-targeted protein, we first integrated AID degron and an NG fluorescent protein into basket nucleoporins' genes and then integrated TIR1-containing cassette into the RCC1 locus.

**Plasmid construction.** The CRISPR/Cas9 system was used for endogenous gene targeting. All gRNA plasmids were generated with primers listed in Supplementary Table 1 (IDT) and integrated into pX330 (Addgene #42230) vector using Zhang Lab General Cloning Protocol[49]. The sequences of homology arms were amplified from genomic DNA extracted from DLD-1 cells, and the full-length 229 amino acid AID (flAID), hygromycin, and TIR1 sequence were amplified by PCR from pcDNA5-EGFP-AID-BubR1 (Addgene #47330)[48] and pBABE TIR1-9Myc (Addgene #47328)[48] plasmids, respectively. The DNA sequence of NG fluorescent protein, FLAG tag, HA tag, a minimal functional AID tag (1×microAID) 71–114 amino-acid[50], and three copies of reduced AID tag (3×miniAID) 65–132 amino-acid[51] were codon optimized and synthesized (IDT). cDNAs of mCherry and puromycin resistance were amplified from pmCherry-N1 (632523, Clontech) and pICE vector (Addgene #46960)[52], respectively.

**Genotyping.** DNA from DLD-1 and CRISPR/Cas9-targeted cells was extracted with the Wizard® Genomic DNA Purification Kit (Promega). Clones were genotyped by PCR for homozygous insertion of tags with two sets of primers listed in Supplementary Table 1.

**Crystal violet staining.** Survival of AID-targeted nucleoporins after auxin treatment was estimated according to Feoktistova[53] with modifications (see Supplementary Methods for details).

**Time-lapse fluorescence microscopy.** Cells were grown on four-well glass bottom chambers (Ibidi), imaged on the Eclipse Ti2 inverted microscope (Nikon), equipped with a spinning disk confocal system (UltraVIEW Vox Rapid Confocal Imager; PerkinElmer), and controlled by the Volocity software (PerkinElmer) utilizing Nikon CFI60 Plan Apochromat Lambda ×60/1.4 oil immersion objective lens with D-C DIC slider ×60 II and ×40/1.3 oil Nikon PlanFluor immersion objective lens. Cells were imaged in FluoroBrite DMEM (ThermoFisher) media. The microscope was equipped with temperature-, $CO_2$-, and humidity-controlled chamber that maintained a 5% $CO_2$ atmosphere and 37 °C.

NG and mCherry fluorescent protein signals were excited with a 488-nm (no more than 20% of power was applied) and 568-nm (no more than 50% of power was applied) laser lines, respectively. A series of 0.5 μm optical sections were acquired every 10 min (Supplementary Fig. 4) or 20 min (Supplementary Fig. 3). Images were captured and analyzed using the Volocity (PerkinElmer) and Image J (National Institutes of Health) softwares, respectively. Images represent maximum intensity projections of entire Z-stacks.

**Immunofluorescence staining.** DLD-1 cells expressing fluorescently targeted nucleoporins were seeded on coverslips in $1.2 \times 10^5$ density on 60-mm dishes and grown for 2 days. Cells were washed with phosphate-buffered saline (PBS), pH 7.4, and immediately fixed with 4% paraformaldehyde in PBS at room temperature (RT) for 15 min, permeabilized with 0.5% Triton X-100 for 10 min, and blocked with 10% horse serum for 20 min. For NUP133 antigen retrieval, cells were washed with PBS and fixed with 4% paraformaldehyde in PBS, containing 0.5% Triton X-100 at RT for 15 min. For NXF1 and GANP NE visualization, cells were propagated on 8-well Nunc Lab-Tek slides for 2 days, then washed three times with Buffer A

(see "NPC-associated proteins by mass spectrometry") and fixed in 4% paraformaldehyde in PBS at RT for 10 min. Localization of other nucleoporins was detected by specific primary antibodies and AlexaFluor-conjugated secondary antibodies (Invitrogen). The nuclei were visualized with Hoechst 33342 (Invitrogen). Images were acquired using an Olympus IX71 inverted microscope (Olympus America Incorporation), equipped with an UltraVIEW spinning disk confocal system (UltraVIEW ERS Rapid Confocal Imager; PerkinElmer), and controlled by the Volocity software (PerkinElmer) utilizing an Olympus UPlanSApo ×100/1.4 oil objective.

Brightness and contrast were applied equally to all images within the experiment using the Fiji software version 2.0.0-rc-68/1.52e. RGB images from Fiji were processed using Adobe Photoshop and Adobe Illustrator CS5.1.

**Analysis of NPC and NPC-associated proteins by mass spectrometry.** NPC-containing fraction (NPC-enriched, NPCCF) was isolated according to Cronshaw[54], with modifications. In brief, cells were incubated 2 × 5 min in buffer A (20 mM HEPES, pH 7.8; 2 mM dithiothreitol (DTT), 10% sucrose, 5 mM $MgCl_2$, 5 mM EGTA, 1% TX100, 0.075% sodium dodecyl sulfate (SDS)) at RT, followed by incubation in buffer B (20 mM HEPES, pH 7.8; 2 mM DTT, 10% sucrose, 0.1 mM $MgCl_2$), containing 4 μg/ml RNase A (Promega) for 10 min at 37 °C, and washed in buffer A. NPCCF was eluted by incubating the plates in buffer C (20 mM HEPES, pH 7.8; 150 mM NaCl, 2 mM DTT, 10% sucrose, 0.3% Empigen BB) for 10 min at 37 °C. The supernatant was spun for 3 min at $28,000 \times g$ (HB-6 rotor) at 4 °C, transferred to the new tubes, incubated on ice for 30 min, and saturated TCA was added to the final concentration of 8%, followed by centrifugation and subsequent washes of the pellet in ethanol. The NPCCF pellet was then solubilized in buffer D (8 M urea, 5 mM DTT) by pipetting, rigorously vortexed, and centrifuged for 1 min at $11,000 \times g$. The supernatant was collected, and the protein concentration was measured.

Total cell lysates were prepared as follows: Cells, grown on 150 mm plate, were washed 5 times with PBS, scraped and pelleted down at $300 \times g$ for 5 min. The pellet was tapped loose and quickly resuspended in 1 ml of lysis buffer (8 M urea, 10 mM DTT) and incubated for 15 min on ice. In all, 0.7 ml of viscous lysate was ultracentrifuged at $500,000 \times g$ (Beckman TLA-120 rotor) for 1 h at +4 °C. The supernatant was stored at −70 °C before processing for mass spectrometry or SDS–polyacrylamide gel electrophoresis (PAGE).

Samples were either individually analyzed with liquid chromatography mass spectrometry (LCMS) for label-free quantitation or analyzed as multiplex sets, after labeling with TMT reagents (TMT10plex label reagent set, Thermo Scientific), according to the manufacturer's instructions. The mass spectrometric analysis was conducted on an LTQ Orbitrap Lumos (Thermo Fisher Scientific) based nanoLCMS system. Protein identification and quantitation analysis were carried out on Proteome Discoverer 2.2 platform (Thermo Fisher Scientific). Peptide IDs were assigned by searching the resulting LCMS raw data against UniProt/SwissProt Human database using the Mascot algorithm (V2.6, Matrix Science Inc.). And peptide-spectrum matches were further validated with Percolate algorithm. We determined the protein-level fold changes based on the median of peptide-level fold changes from the Proteome Discoverer-produced abundances in handling both TMT and Label-free results. TMT- and Label-free fold changes of relevant proteins for NPC-enriched and total cell lysate are shown in Supplementary Data 1 and full protocol is described in Supplementary Methods.

**Fluorescence loss in photobleaching.** DLD-1 $^{AID}$TPR-NXF1$^{mCherry}$ cells were seeded on 4-well glass-bottom chambers (Ibidi) in FluoroBrite DMEM supplemented with heat-inactivated 10% FBS (Atlanta Biologicals) and antibiotics (100 IU/ml penicillin and 100 μg/ml streptomycin). Cells were imaged on a Zeiss LSM 780 confocal microscope with a ×63/1.4 Plan-Apochromat oil objective, with a stage-top incubator set at 37 °C and 5% $CO_2$. Fluorescence images were recorded in the red channel at zoom ×5, 256 × 256 pixels, with the excitation set at 1% of the 561 nm laser and the pinhole fully open. Differential interference contrast images were recorded in parallel to better identify the nucleus and NE. For FLIP, we used a 120-point time series with an interval of 0.195 s. Five prebleached images of each cell were taken, and then the center of the cell nuclei were bleached by 100% 561 nm laser in an area of 44 × 39 pixels (4.6 × 4.1 μm) with 20 iterations. Bleaching was repeated every image, increasing the delay between images to 0.840 s.

The intensity of the bleached region of interest (ROI) and the NE (npcROI, 6 × 12 pixels or 0.63 × 1.26 μm) were measured every time point, together with the intensity of the NE signal of the neighboring cell (cROI) to assess fluorescence losses due to the imaging laser. The relative fluorescence intensity of the prebleached region was taken as 100%. To calculate normalized npcROI (Fn), we used the following equation $Fn = \frac{(npcROI\ ti/r\ ti)}{(npcROI\ t0/r\ t0)} *100$, where npcROI $t0/r$ t0 is the normalized fluorescent intensity at time 0 and npcROI ti/$r$ ti at the given time point, and $r$ is the coefficient of photobleaching rate. To quantify photobleaching rate, we used cROI of neighboring control cells before (cROI t0) and after (cROI ti) photobleaching and the following equation $r = \frac{cROI\ ti}{cROI\ t0}$. The data collected after photobleaching were plotted using the Prism software and compared using unpaired $t$ test.

**Protein transport assay**. The nuclear transport assay was adapted from the protocol outlined in Niopek et al.[55]. Briefly, [AID]TPR cell lines were seeded on Ibidi μ-Slide glass bottom slides, transfected with 500 ng of NLS-mCherry-LEXY plasmid (pDN122) (Addgene plasmid # 72655[55]), and kept in the dark for 24 h prior to imaging. NLS-mCherry-LEXY positive cells were first excited with a 561 nm laser line with 30 ms exposure and imaged every 30 s for 10 min. Next, the cells were exposed to 405 nm laser line with 1 s exposure every 30 s to induce nuclear export of model substrate, which was monitored for 15 min. The 405-nm laser was then shut off to induce nuclear import of model substrate, which was monitored for 20 min. During the course of the experiment, cells were imaged every 30 s using 561 nm laser to follow mCherry signal of the model substrate. Image analysis was performed on the Volocity (PerkinElmer) and ImageJ (National Institutes of Health) softwares with Time Series Analyzer V3 plugin and ROI Manager dialog box.

**Protein extraction and Western blot**. Pellets of DLD-1 cells were lysed in 2× Laemmli Sample Buffer (Bio-Rad), boiled for 15 min at 98 °C, and ultra-centrifuged at $500,000 \times g$ (TLA-120.1 rotor) for 10 min at 16 °C. SDS-PAGE and western blotting were performed as described elsewhere[56], see Supplementary Methods for details.

**Quantitative PCR with reverse transcription**. Trizol-extracted RNA from cells was treated with Turbo DNA-free DNAse (ThermoFisher) and purified on the RNAeasy MiniKit (Qiagen) columns according to the manufacturer's protocol. One thousand three hundred micrograms of DNA-free RNA were used for cDNA synthesis with random hexamers (NEB) using the SuperScript II (ThermoFisher) Kit as suggested by the protocol. Quantitative PCR was performed using Prime-Time® Predesigned qPCR Assays (IDT, Supplementary Table 1) using CFX Connect™ Real-Time PCR Detection System (Bio-Rad). The relative transcript level was determined by normalizing to the expression level of ACTB and POLR2A genes using ddCq-method. Relative expression of differentially expressed genes were plotted with scatter plot-bar function using the Prism software. Data are presented as the mean value where error bars are SD. Unpaired two-tailed Student's $t$ test has been applied for comparison of untreated and auxin-treated cells.

**RNA-seq sample preparation**. AID-targeted cells at 3rd–6th passage were seeded on 6-well dishes. A million of adherent DLD-1 cells were washed with PBS and lysed directly with RLT buffer from the RNAeasy Mini Kit (Qiagen). To inhibit CRM1, [AID]TPR cells were treated with 200 nM of Leptomycin B (LCLabs) for 3 h. Extracted RNA was analyzed for 260/280 and RNA integrity number (RIN) value was assessed on Bioanalyzer 2100 (Agilent); only samples with RIN value >9 were proceeded for the library construction using the Illumina TruSeq Stranded Total RNA Library Preparation Kit (Illumina) according to the manufacturer's instructions. All RNA samples were depleted for ribosomal RNA before library construction using the Ribo-Zero™ Gold Kit H/M/R Kit (Illumina). Hi-Seq run of three independent biological replicates was performed on Illumina HiSeq2500. Twenty-thirty millions of 101 bp paired-end reads were generated per each replicate.

**RNA-seq analysis**. We aligned the short reads to the reference human genome GRCh38.p7/hg38 that we obtained from Ensembl[57]. We used STAR aligner version 2.5.2b[58] for the mapping with default parameters. We quantified gene expression levels using featureCounts 1.4.6[59]. To obtain gene-level read counts, we supplied a gene annotation model from Ensemble Release 87, which corresponds to GEN-CODE version 25. We calculated gene-level fragments per kilobase of transcript per million mapped read (FPKM) values based on the longest possible exon lengths after collapsing overlapping exons. For DE analysis, we used DESeq2 1.18.1[60] as in the reference with one minor change; we applied expression cutoffs FPKM > 1 and counts per million >1 before the analysis. We studied and visualized genes whose expression levels are above the cutoff from >5% of our samples excluding for the ActD-treated samples. We used $k$-means clustering function in R environment 3.3.2 (R Code Team, 2018) to generate heat maps in the "Result" section.

The Venn diagrams were created using Venny 2.1 tool and BioVinci software version 1.1.5, r20181005. GO term enrichment analysis for BSK-NUPs was performed using an integrated HumanMine database[61] V5.1 2018 with Holm–Bonferroni test correction ($p$ value 0.05) for Biological Processes.

**SLAM-seq sample preparation**. SLAM-seq of nascent RNAs was performed with $S^4U$ incorporation using the Anabolic (061.24) Kinetics Module Kit purchased from Lexogen company. Briefly, before labeling of nascent RNA, each well of [AID]-TPR cells from the 6-well plate was replaced with fresh media or media containing 1 mM auxin for 2 h. For chasing, cells were treated with media containing 300 μM $S^4U$ or 300 μM $S^4U$ supplemented with 1 mM auxin for 0 min, 30 min, 60 min, or 2 h. All operations with $S^4U$ were performed under a red light source to protect $S^4U$ from crosslinking. Total RNA from triplicates was collected in 1 ml of TRIzol Reagent (Invitrogen), stored at −80 °C, and further purified according to Lexogen manufacturing protocol. Five micrograms of total RNA were alkylated with freshly prepared 100 mM iodoacetamide. RNA quality and concentration were measured using the Qubit 4 RNA IQ and RNA BR Assay Kit (ThermoFisher Scientific). QuantSeq 3′ mRNA-seq Libraries were sequenced, and a total of 8–24 million reads

were generated per sample. The SLAM-seq data were processed with a SLAM-DUNK pipeline[62]. The reads were aligned to the human genome GRCh38.p10. DE analysis of TcReadCounts was performed with DESeq2 (version 1.26.0) using R 3.5.1, adj. $p$ was FDR adjusted. For global normalization, sizeFactors were used based on total ReadCounts. For gene patterns analysis, we used degPatterns from the package DEGreport 1.22.0 with the default options except for minc = 2 and reduce = TRUE. Normalized TcReadCounts data have been transformed by VST (variance stabilization transformation), thereby the variability of the values was not related to their mean, allowing comparison of the gene patterns. In particular, the clustering of DE genes for gene patterns was computed using the normalized log2 T > C counts. The mean normalized log2 T > C counts were then centered to the mean and scaled to the standard deviation to obtain $Z$-scores and plotted in boxplots. $Z$-scores were calculated per gene after clustering. The boxplot minima corresponds to the smallest observations greater than or equal to lower bound − $1.5 \times$ IQR (interquartile range), the lower bound corresponds to the 25% quartile, center to the 50% quartile, upper bound to the 75% quartile, and the maxima to the largest observation less than or equal to upper bound + $1.5 \times$ IQR. Gene patterns were plotted in Prism 5.0b. To prepare M-A plots, we used genes with adj. $p < 0.1$ and log2FoldChange >0.

**ChIP assay with sequencing (ChIP-seq) analysis**. ChIP-seq was performed using modified ChIP and ChIP-seq protocols previously described[63,64]. Briefly, $3.5 \times 10^6$ cells per 100 mm dish were washed with PBS and cross-linked with 1% formaldehyde for 11 min. After quenching with 0.125 M glycine for 13 min, cells were scraped, lysed, and sheared by sonication using Bioruptor (Diagenode) for 42 min with 30 s pulse/pause cycles on ice. After enrichment of chromatin fragments around 300-bp (as assayed by agarose gel electrophoresis), cell debris was spun down, and supernatant was incubated overnight at +4 °C with anti-mouse IgG M-280 (ThermoFisher) Dynabeads pre-bound with anti-Ser5P Pol II mouse antibodies (ab5408) for 10 h. Immune complexes were washed twice with IP, IP-500, LiCl, and TE buffers and eluted using the CHIP Elute Kit (Clontech). Two independent biological replicates were used for library construction using the DNA SMART ChIP-Seq Kit according to the manufacturer's protocol. In total, 16 PCR cycles were used during library construction. Two hundred million 50-bp paired-end reads were generated for combined replicates. Alignment of short reads was performed with BWA (VN:0.7.17-r1188) against reference human genome GRCh38.p7/hg38. Read quantitation over genes was tabulated using HOMER[65]. Differential IP recovery of genes was tested using DESeq2[60] and MACS2 broad peak calling tools. Plots were generated with the IGV_2.4.6 software.

**RNA FISH**. poly(A)RNA FISH was performed as described earlier[66]. In brief, cells were treated with 1 mM auxin for 8 h, washed with PBS, fixed for 15 min in 4% PFA at RT, incubated in 70% ethanol for 16 h at 4 °C, washed with PBS, and incubated with wash buffer for 5 min at RT. Cells were then incubated with oligo (dT)-Quasar 670 probes (LGC Biosearch Technologies) for 4 h at 37 °C, followed by incubation with wash buffer at 37 °C for 30 min and then washed with PBS. Cells were then stained with 1.5 μg/ml Hoechst 33258 (Molecular Probes/Life Technologies) for 10 min, washed with PBS, and finally mounted in ProLong Gold antifade reagent (Life Technologies).

Individual RNAs were detected by FISH using ViewRNA Cell Plus (Thermo Fisher Scientific, Cat#88-19000-99) and corresponding RNA probes (Thermo Fisher Scientific, Cat# VX-01), according to the manufacturer's instructions. Briefly, cells were seeded on 24-well plates with coverslips, processed once they reach 80% confluency, and fixed according to the ViewRNA Cell Plus protocol. Mab414 antibodies were used to detect NPC.

Cytoplasmic-to-nucleus (Cyt/Nu) ratio of mRNA foci was plotted with scatter plot function in Prism software. Data are expressed as mean value with standard deviations as error bars. Unpaired two-tailed Student's $t$ test has been applied for comparison of untreated and auxin-treated cells.

**RNA FISH microscopy and data analysis**. poly(A)RNA FISH image capturing and analysis was performed as described earlier[66]. Images were captured with a Zeiss Axiovert 200 M automated microscope using the AxioVision software. A Zeiss ×60 Plan-APOCHROMAT lens (1.4 NA) and Zeiss AxioCam MRm camera were used in imaging. Multiple 0.3 μm $Z$ planes were captured and images were deconvolved with the AutoQuant software. Imaris Surfaces tool was used for segmentation and signal analysis within the cytoplasm and nucleus. Images are representative of 18 images from two independent experiments. The graph provides the ratio of cytoplasmic to nucleus (C/N) poly(A)RNA with and without auxin treatment. Values represent means ± SD of at least 25 cells from biological triplicates. Images are representative of nine images from biological triplicates.

Individual RNAs detected with ViewRNA Cell Plus were imaged and analyzed on Olympus IX71 inverted microscope utilizing an Olympus UPlanSApo ×60/1.4 oil objective as described in "Immunofluorescence staining" section.

Cyt/Nu ratio and the total number of mRNA foci were plotted with scatter plot-bar function using the Prism software. Data are presented as the mean value where error bars are SD. Unpaired two-tailed Student's $t$ test has been applied for comparison of untreated and auxin-treated cells.

**Antibodies**. The following antibodies were used for immunostaining and western blot analysis: anti-mouse (A28175) and anti-rabbit (A11034) AlexaFluor-488-conjugated antibodies (Invitrogen), anti-mouse (A11004) and anti-rabbit (A11011) AlexaFluor-568-conjugated antibodies (Invitrogen), anti-mouse (A27042) AlexaFluor-680-conjugated antibodies (Invitrogen), and anti-mouse (A-31553) AlexaFluor-405-conjugated antibodies. Specific primary antibodies against NUP50 (A301-782A), NUP153 (A301-789A), TPR (A300-828A, A301-827A), GANP (A303-128A for WB; A303-127A for IF), NXF1 (A303-913A for IF; A303-915A for WB), NUP133 (A302-386A), NUP98 (sc-30112), HA (11867423001), FLAG M2 (F1804), ACTB (13E5, 4970S), SON (GTX129778), and Mab414 (MMS-120P) were purchased from Bethyl Laboratories, Santa Cruz Biotechnology, Sigma-Aldrich, GeneTex, and BioLegend, respectively. Secondary horseradish peroxidase-conjugated anti-mouse and anti-rabbit antibodies for western blot analysis were purchased from Sigma-Aldrich.

**Statistics and reproducibility**. Both immunofluorescence and live imaging experiments were performed at least three times. FISH experiments were repeated twice by two independent investigators. For quantification of imaging experiments, we analyzed three independent fields. qRT-PCR data are the result of one representative experiment repeated at least three times. RNA-seq and SLAM-seq data were analyzed from three independent biological replicates. ChIP-seq data represent two independent biological replicates. If not indicated in the figure legend, information about the number of analyzed fields or cells is provided in the Source data file.

**Reporting summary**. Further information on research design is available in the Nature Research Reporting Summary linked to this article.

## Data availability

The sequencing data that support the findings of this study have been deposited in NCBI Gene Expression Omnibus (GEO, http://www.ncbi.nlm.nih.gov/geo/) with the accession codes ID GSE132363 (RNA-seq), GSE148560 (RNA-seq, CRM1), GSE149890 (SLAM-seq), and Sequence Read Archive (SRA) under BioProject ID: PRJNA548782 (ChIP Pol II Ser5P). The mass spectrometric proteomics data have been deposited to the ProteomeXchange Consortium via the PRIDE[67] partner repository with the dataset identifier PXD020075. All other data, including immunofluorescence images supporting the finding of this study, are available within the article, Supplementary Figures, and will be provided by the authors upon reasonable request. Source data are provided with this paper.

## Code availability

Standard published open-source tools were used (see "Methods"); no algorithms or tools were developed for this work.

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

## Acknowledgements
V.A., A.A., S.C., M.D., C.E., H.L., K.C.Y., C.E., and A.S. were supported by the Intramural Research Program of the *Eunice Kennedy Shriver* National Institute of Child Health and Human Development at the National Institutes of Health, USA (Intramural Project #Z01 HD008954). P.B. and B.F. were supported by NIH R01 GM113874-04, NIH 1 R01 AI154635, and R01AI125524-04. We are indebted to Y. Chen and M. Gucek (NHLBI Proteomics Core) for help with mass spectrometric analysis. We grateful to T. Li (NICHD Molecular Genomics Core) and S. Herafeld and R. Dale (Bioinformatics and Scientific Programming Core) for help with the library construction, paired-end sequencing, and bioinformatics support. We thank V. Schram (NICHD Microscopy & Imaging Core) for assistance with FLIP experiments. We also thank Brian Brown, NIH Library Editing Service for reviewing the manuscript.

## Author contributions
V.A., A.A., and M.D. developed hypothesis, conceived, and designed the analysis and wrote the paper. V.A. collected data and developed the project. S.C., C.E., V.A., R.K., A.S., P.B., K.C.Y., and A.A. carried out the experiments. H.L., J.I., and C.E. performed bioinformatics analysis. H.L., A.S., S.C., P.B., B.F., and C.E. edited the manuscript.

## Competing interests
The authors declare no competing interests.
