## [Peer Review File · Nature Communications]

Reviewers' comments:

Reviewer #1 (Remarks to the Author):

The manuscript by Aksenova et al. investigates the role of nuclear pore complex (NPC) components (nucleoporins) in nuclear transport and other NPC-dependent processes using an innovative experimental strategy, i.e. the rapid and specific depletion of individual nucleoporins through auxin-induced degradation. By combining this approach with microscopy analyses and proteomics, the authors first assess the interdependency between nuclear basket Nups (Nup153, Nup50, Tpr) for their association with NPCs. Their study demonstrates that Nup153 is required for the recruitment of Nup50 at NPCs (in agreement with published reports; PMID:12802065, 23007389), but dispensable for the NPC localization of Tpr, in contrast with previous findings (PMID:12802065, 23591820). This work further reveals unique phenotypic signatures associated with individual Nup depletions, with respect to nuclear transport (protein import, protein export, poly-A+ RNA export) and gene expression. Strikingly, the transcriptomic profile associated with Tpr depletion resembles the one observed upon inactivation of TREX-2, an mRNA export factor previously reported to associate with the NPC basket in a Tpr/Nup153-dependent manner (PMID:23591820). Consistently, Tpr depletion is also shown here to impact both TREX-2 association to NPCs and poly-A+ RNA export.

Overall, this is a well-designed study which describes an unprecedented methodology to evaluate the individual function of nucleoporins, avoiding the indirect effects likely caused by extended NPC disruption in other assays. In this frame, the tools and methodology presented in this manuscript will be extremely useful for the field. However, this work only brings little additional insights into the function of nuclear basket nucleoporins and the mechanisms by which they influence gene expression.

Specific comments

1. It is shown that Nup153 depletion compromises the post-mitotic recruitment of Tpr at NPCs, but does not affect its localization in interphase, conflicting with previous reports using siRNAs to inactivate Nup153.

- it seems that Nup153 depletion causes a mild yet reproducible reduction in Tpr levels in NPC mass spectrometry analyses (Suppl. Table 1), as well as decreased Tpr signals at the NE in immunofluorescence quantifications (Suppl. Fig. 3d), suggesting that Nup153 also contributes to some extent to the stable association of Tpr with NPCs in interphase. Can this be confirmed/quantified by immunoblot experiments and further discussed?

- could the authors indicate how they distinguish post-mitotic cells in Suppl. Fig. 4e,f?

- the authors state that "Tpr ... fails to be imported to the nucleus in post-mitotic Nup153-depleted cells." However, in this situation, Tpr form foci in the cytoplasm and the nucleus (Suppl. Fig. 4e), suggesting that Tpr import occurs. Can the conclusion be rephrased accordingly?

2. The authors report that Tpr depletion triggers changes in the gene expression profile, but do not identify the underlying mechanisms: they should determine whether Tpr depletion primarily impacts mRNA synthesis or degradation.

- to investigate transcription rates, the authors have analyzed Pol II density/distribution by ChIP-seq using a CTD-Ser5P-specific antibody. However, Ser5 phosphorylation is not the most accurate readout to score changes in transcription (see PMID: 26844429). ChIP experiments using CTD-Ser2P- or NTD-specific antibodies have to be performed to analyze the impact of Tpr depletion on gene transcription. In addition, the authors should distinguish up- and down-regulated genes when they characterize the overlap between the genes exhibiting changes in RNA levels and those with changes in Pol II recruitment (Fig. 4b).

- if these experiments confirm that most Tpr-dependent mRNAs have unchanged transcription rates in Tpr-depleted cells, the authors should examine their stability and degradation. This could be easily performed by analyzing the kinetics of mRNA degradation upon transcription inhibition for a subset of representative target genes.

- the results obtained following Actinomycin D treatment have to be taken with caution. Besides its

impact on mRNA synthesis, ActD can also interfere with mRNA stability (PMID: 19111184). Indirect effects are thereby likely to explain the discrepancy between the ActD-sensitivity scored for most Tpr-induced transcriptomic changes (Fig. 4a) and the apparent lack of effect of Tpr depletion on transcription (Fig. 4b).

3. The functional relationship identified between Tpr, GANP (TREX-2) and NXF1 raises additional questions:

- Tpr depletion affects GANP association to NPCs but does not disturb NXF1 recruitment (Fig. 3c,d). Yet, Tpr, GANP and NXF1 share specific transcriptomic signatures (Fig. 2d). One explanation could be that Tpr depletion affects NXF1 dynamic association with NPCs without detectable impact on its steady-state NPC levels as scored in fixed cells. Could the authors check NXF1 dynamic association with NPCs by performing FRAP in Tpr-depleted cells (see PMID:23591820)?
- while Tpr depletion leads to a strong defect in GANP association with NPCs, it only affects a subset of GANP target mRNAs (Fig. 3c and 2d). Can the authors comment on this apparent discrepancy?

Minor remark

In some figures showing means \pm -SD, the number of replicates (n=) has to be indicated.

Reviewer #2 (Remarks to the Author):

Aksenova et al. report findings on the structural and functional specialization of the NPC basket, an important substructure of the NPC. Using an auxin-induced-degron system to rapidly deplete basket nucleoporins (Nups), they found that the loss of individual Nups caused distinct transcriptome changes in a human DLD1 cell line. TPR, the largest basket Nup, was found to be required for localization of GANP, a TREX-2 subunit, to the NPC. Moreover, TPR and GANP show overlapping transcriptome changes after depletion suggesting that they could function somewhere along the same gene expression pathway, which is distinct from pathways influenced by the other basket Nups, NUP153 or NUP50.

Overall, the structural analysis of the NPC basket has lagged behind and there is a great need to fill our gaps of knowledge. There are also conflicting reports about how basket Nups are recruited to the NPC core and the differences in the literature may stem, in part, from the methods that were used for Nup depletion (the authors have correctly cited the literature). One possible reason is that RNAi methods are often inefficient and require lengthy incubations, which could give rise to secondary phenotypes. Here, AID is a particularly effective and elegant method that is clearly superior to previous approaches. Overall, this paper is well-written and technically sound. The Nup localization experiments are very convincing. I think that the paper is potentially interesting for the field. Having said that, I would have liked to see a deeper exploration of the TPR / TREX-2 effects on gene expression. The lack of mechanistic data somewhat limits the appeal and novelty of the paper. Key open questions are how TREX-2/TPR synergize to regulate gene expression, how these proteins specifically recognize their target genes and which factors connect TREX-2 to other parts of the gene expression machinery.

Major criticisms:

1) Compared to yeast TREX-2, the mammalian TREX-2 complex is structurally and functionally poorly characterized. I would like to see whether the other TREX-2 subunits PCID2, ENY2 and centrin-2/3 also become mislocalized upon TPR depletion by AID. This would strengthen the paper and should be doable.

2) Transcriptome changes are a rather poor functional readout because steady-state mRNA

measurements are influenced by many factors. TPR loss appears to specifically impact RNAs, which are involved in regulation of transcription. This is potentially interesting, but what does it mean? The authors should follow-up on this observation.

Similarly, I don't see how the authors can conclude from comparing Ser5 ChIP-Seq data with mRNA levels that *fjx1* is regulated at the post-transcriptional level. Overall, the authors should tone down their conclusions about whether or not transcription rates are affected by loss of TPR / GANP. Methods for detecting nascent, actively transcribed Pol II transcripts are necessary to reach solid conclusion. This is beyond the scope of the paper, hence, a textual revision is appropriate.

3) What is the functional relationship between transcripts that are misregulated upon TPR/GANP depletion and those that are not properly exported in these cells? Are these distinct or overlapping sets of transcripts?

4) Have the authors tried to ChIP-Seq TPR and GANP? This would be a major asset for the paper.

Minor comments:

1) The authors should better discuss why their data on the basket Nup targeting to the NPC core differs from earlier work. I still find the data confusing, particularly in comparison to the Umlauf et al., 2013 paper.

Reviewer #1

This reviewer comments that “Overall, this is a well-designed study which describes an unprecedented methodology to evaluate the individual function of nucleoporins, avoiding the indirect effects likely caused by extended NPC disruption in other assays.” The reviewer also noted that the methodology that we have developed “will be extremely useful for the field”. The main issue raised by the reviewer was a desire for more mechanistic insight regarding how Tpr influences gene expression.

We appreciate the supportive comments of the reviewer and have worked extensively to address this major issue. We now provide additional data showing that TREX-2 component ENY2 and PICD2 depend upon on Tpr for NPC association (Figure 4, Supplementary Figure 7b) and we have extensively investigated the origin of transcriptomic changes after Tpr loss, demonstrating changes in both the rates of transcription and export (Figures 5, 6). These data further strengthen our conclusion that Tpr acts as an integral component of the TREX-2 RNA export pathway and that it influences RNA synthesis and export in multiple ways.

Our detailed responses to the reviewer’s points are as follows:

Point 1a: It is shown that Nup153 depletion compromises the post-mitotic recruitment of Tpr at NPCs, but does not affect its localization in interphase, conflicting with previous reports using siRNAs to inactivate Nup153.

it seems that Nup153 depletion causes a mild yet reproducible reduction in Tpr levels in NPC mass spectrometry analyses (Suppl. Table 1), as well as decreased Tpr signals at the NE in immunofluorescence quantifications (Suppl. Fig. 3d), suggesting that Nup153 also contributes to some extent to the stable association of Tpr with NPCs in interphase. Can this be confirmed/quantified by immunoblot experiments and further discussed?

To address questions about Tpr dependence upon Nup153 in the manner that the reviewer requests, we performed Western blot analysis of the basket nucleoporins abundance in Tpr-, Nup153-, and Nup50-depleted cells using both NPC-enriched nuclear extracts (to assess the loss of nucleoporins from the nuclear envelope) and total cell lysates, and now show that the loss of Nup153 did not visibly affect Tpr amount (Supplementary Fig. 3d). Of note, samples for Western blotting were prepared in the same way as those for mass spectrometry (Suppl. Table 1).

We believe that mild reduction (about 25%) of Tpr abundance after Nup153 loss that we observed in ultra-sensitive TMT analysis can be explained by the source of material: we performed this mass-spectrometry analysis using asynchronous cells. Because recruitment of Tpr to NPC in post-mitotic cells requires Nup153, the inclusion of such cells (~ 10% during 2 h auxin treatment. Duration of cell cycle in DLD1 cells ~ 20 h) in our analysis should result in ~ 10% reduction of Tpr in Nup153 Auxin+ samples. In addition, it is also possible that Nup153 is required for the recruitment of Tpr during de novo formation of NPC in interphase. A slight reduction in mCherry signal (Supplementary Fig. 4d) could be explained by both photobleaching (we also noticed mild photobleaching in control cells at 60 min time point) and the putative defect in the de novo NPC formation. Per reviewer’s suggestion, we discuss these possibilities in the text (p.6, p.12)

Point 1b: Could the authors indicate how they distinguish post-mitotic cells in Suppl. Fig. 4e,f?

We followed mitotic cell division using live imaging of cells by tracking endogenously tagged RCC1-iRFP. Because RCC1 binds both Histones H2A/B and DNA, it can be used to faithfully monitor chromatin dynamics at all stages of the cell cycle (new Supplementary Fig. 3e-f).

Point 1c: The authors state that “Tpr ... fails to be imported to the nucleus in post-mitotic Nup153-depleted cells.” However, in this situation, Tpr form foci in the cytoplasm and the nucleus (Suppl. Fig. 4e), suggesting that Tpr import occurs. Can the conclusion be rephrased accordingly?

We prepared new images (Supplementary Fig.3e-f) from the live imaging experiment of cells, released from the G2/M block. Tpr-mCherry co-localizes with Nup153-NG-AID all the time in auxin-free media and auxin-treated cells until nuclear envelope breakdown. Post-mitotic Nup153-depleted cells were characterized by the restricted nuclear growth. Importantly, aggregates of Tpr-mCherry reside in the cytoplasm, indicating that nuclear import of Tpr does not occur. The “presence” of Tpr aggregates in the image that we used in our original manuscript was due to the inclusion of the cytosolic Tpr aggregates that were located on top of the nucleus (as we combined stacks of images, 1 μm distance, 10 μm total, into one image).

Point 2a: The authors report that Tpr depletion triggers changes in the gene expression profile, but do not identify the underlying mechanisms: they should determine whether Tpr depletion primarily impacts mRNA synthesis or degradation.

To investigate transcription rates, the authors have analyzed Pol II density/distribution by ChIP-seq using a CTD-Ser5P-specific antibody. However, Ser5 phosphorylation is not the most accurate readout to score changes in transcription (see PMID: 26844429). ChIP experiments using CTD-Ser2P- or NTD-specific antibodies have to be performed to analyze the impact of Tpr depletion on gene transcription.

We attempted to perform CTD-Ser2P ChIP-sequencing experiment using Ser2P specific antibody (ab5095 and MABE953, PMID: 22011111) Unfortunately, ChIP Ser2P signal did not cover the coding regions of the genes, as expected, but only showed sharp peaks around 3'UTRs. Because the data differed from the reported observations, we decided not to include this result in the manuscript. However, we performed a S⁴U-based SLAM-sequencing experiment that allowed us to follow nascent transcription. The results indicate that gene expression changes observed after the loss of Tpr could be explained by the changes in transcription rates of corresponding genes (new Figures 5 and 6).

Point 2b: In addition, the authors should distinguish up- and down-regulated genes when they characterize the overlap between the genes exhibiting changes in RNA levels and those with changes in Pol II recruitment (Fig. 4b).

We followed reviewer's advice and split the up- and down-regulated genes and compared them separately to Ser5P Pol 2 ChIP-sequencing data (Supplementary Fig. 9e).

Point 2c: If these experiments confirm that most Tpr-dependent mRNAs have unchanged transcription rates in Tpr-depleted cells, the authors should examine their stability and degradation.

The data obtained from SLAM sequencing suggest that loss of Tpr affects the synthesis rate of majority (116) of Tpr-dependent mRNAs (new Figures 5 and 6). In addition, we attempted to determine mRNA stability upon Tpr loss by following a SLAM-seq “catabolic” protocol (please see "Supplementary Figure for the reviewers"). Unfortunately, prolonged (24 h) labelling with 100 μM S⁴U, as was suggested by the company (Lexogen), suppressed transcription of many genes in DLD-1 cells. Therefore, removal of S⁴U (to initiate the analysis of stability of S⁴U-labelled transcripts) apparently resulted in rapid accumulation of remaining S⁴U that cells had already absorbed from the media, thus substantially confounding interpretations of the results.

Considering these issues, we decided not to include these data into the manuscript but would like to address the question of co-transcriptional stabilization or degradation of newly synthesized transcripts in a

separate paper. However, initial results from a much simpler analysis (kinetics of mRNA degradation after transcription inhibition) suggest that Tpr does not substantially affect mRNA stability within 2 hours window (see "Supplementary Figure for the reviewers, panel j").

Point 2d: This could be easily performed by analyzing the kinetics of mRNA degradation upon transcription inhibition for a subset of representative target genes.

The results obtained following Actinomycin D treatment have to be taken with caution. Besides its impact on mRNA synthesis, ActD can also interfere with mRNA stability (PMID: 19111184). Indirect effects are thereby likely to explain the discrepancy between the ActD-sensitivity scored for most Tpr-induced transcriptomic changes (Fig. 4a) and the apparent lack of effect of Tpr depletion on transcription (Fig. 4b).

We acknowledge the point raised by the reviewer. Actinomycin D impacts mRNA synthesis, interferes with mRNA stability and is a significant stress for the cell (PMID: 21922053). Moreover, 5 µg/ml Actinomycin D concentration can inhibit RNAP II and result in aggregation of several proteins from the nucleoplasm into nucleolar caps (PMID: 15758027).

We performed RT-qPCR analysis on a subset of transcripts (please see "Supplementary Figure for the reviewers, panel j"). RNA abundance and kinetics of mRNA degradation for a subset of representative target genes were analyzed in AID-Tpr cells after 0h, 2h, 4h, 6h, 8h of ActD or ActD+auxin treatment. We did not find significant changes in RNA abundance between control and Tpr-depleted cells within the 2h window, suggesting that RNA degradation might not be affected by Tpr loss. However, we decided not to include these data into the manuscript because substantiating involvement of Tpr in co-transcriptional stabilization or degradation of newly synthesized transcripts would require a large number of additional experiments.

Point 3: The functional relationship identified between Tpr, GANP (TREX-2) and NXF1 raises additional questions:

- Tpr depletion affects GANP association to NPCs but does not disturb NXF1 recruitment (Fig. 3c,d). Yet, Tpr, GANP and NXF1 share specific transcriptomic signatures (Fig. 2d). One explanation could be that Tpr depletion affects NXF1 dynamic association with NPCs without detectable impact on its steady-state NPC levels as scored in fixed cells.
- Could the authors check NXF1 dynamic association with NPCs by performing FRAP in Tpr-depleted cells (see PMID:23591820)?

We performed the experiment, as suggested by the reviewer. We tagged endogenous NXF1 with mCherry and followed its dynamics. Interestingly, in contrast to previously reported NE localization of overexpressed GFP-tagged NXF1, we observed that the majority of mCherry-tagged NXF1 is localized in the nucleoplasm when expressed at endogenous levels, and the NXF1 signal at the NE was difficult to visualize against this nucleoplasmic background (Supplementary Fig. 8a). Under these circumstances, NPC-bound NXF1 could be visualized only after pre-wash of live cells with detergent-containing buffer that released nucleoplasmic NXF1. In particular, we used the same buffer to prepare NPC-enriched samples for mass-spectrometry analysis and for NXF1 immunohistochemistry (Figure 3 c-g).

The dominant nucleoplasmic signal made it impossible to perform FRAP experiments to analyze NXF1 association to the NE in live cells. However, we performed a FLIP experiment (Supplementary Fig. 8b-c) to assess the dynamics of exchange between NPC-bound and nucleoplasmic NXF1 after Tpr loss. In this experiment, the center of the nucleus was continuously bleached and NXF1 fluorescence intensity was measured at the vicinity of NE (as defined by DIC imaging). We could not detect a difference between control and Tpr-depleted cells in NXF1 dynamics at the NE.

Interestingly, Dr. Yaron Shav-Tal's group demonstrated that initial mRNP binding to the NPC does not require NXF1 at the NPC while our manuscript was under revision. They also found that NXF1 is not adjacent to Tpr, but rather occupies positions at the cytoplasmic side of the NPC (PMID: 31375530). Our data suggest that both NXF1-NPC localization (potentially at the cytoplasmic side) and NXF1 dynamics at the vicinity of NE are not affected upon Tpr loss. We speculate that Tpr loss can partially phenocopy loss of NXF1 because NXF1 lies upstream of both Tpr and GANP and could be involved at early steps of mRNA processing (see Discussion, p.13). Interaction of GANP and NXF1 was demonstrated earlier by Dr. Ronald A. Laskey (PMID: 20005110).

Point 4: While Tpr depletion leads to a strong defect in GANP association with NPCs, it only affects a subset of GANP target mRNAs (Fig. 3c and 2d). Can the authors comment on this apparent discrepancy?

GANP protein is localized both at the NPC and within the nucleus interior (based on our observations from PFA-fixed samples). Moreover, GANP binds and recruits NXF1-processing mRNPs particles in the nuclear interior and delivers them to the NPC (PMID:20005110). We show that there is a subset of GANP dependent transcripts that are affected only by GANP but not Tpr loss. We analyzed a subset of GANP-specific (that is, Tpr-independent) mRNAs and found that these transcripts have unique functions that are not shared with Tpr and NXF1-specific transcripts. These transcripts are involved in chromosome organization, DNA packing, and nucleosome assembly and most of them encode histones (Supplementary Fig. 7f), suggesting that GANP may have a special function in processing these mRNAs.

Reviewer #2

The reviewer comments that “AID is a particularly effective and elegant method that is clearly superior to previous approaches. Overall, this paper is well-written and technically sound.” While noting that the manuscript may be of interest to the field, the reviewer also asks for a “deeper exploration of the TPR / TREX-2 effects on gene expression”. This issue is similar to the major concern raised by Reviewer #1.

We appreciate the positive and constructive comments of the reviewer and have worked extensively to address this issue, adding data as noted in the summary comments for Reviewer #1. Our data collectively show that Tpr works in concert with the TREX-2 complex, potentially altering gene expression at multiple points for individual target genes. We believe that the added data provide a much more nuanced picture of Tpr and TREX-2 function, significantly strengthening the manuscript.

Our detailed responses to the reviewer’s points are as follows:

Point 1: Compared to yeast TREX-2, the mammalian TREX-2 complex is structurally and functionally poorly characterized. I would like to see whether the other TREX-2 subunits PCID2, ENY2 and centrin-2/3 also become mislocalized upon TPR depletion by AID. This would strengthen the paper and should be doable.

We tested several commercial antibodies to analyze the localization of TREX-2 subunits. In particular, we tested antibodies against PCID2 (Sigma, #HPA 073074), ENY2 (Proteintech, #15778-1-AP), and CENT2 (Proteintech, #15877-1-AP). Unfortunately, all of these antibodies failed to detect the corresponding proteins at the NPC. To overcome this problem, we used CRISPR/Cas9 to endogenously tag PCID2, ENY2, and CETN2 with mCherry and followed TREX2 subunits localization in live cells. mCherry-tagged PCID2 and ENY2 were readily detectable at the nuclear envelope. As expected, both PCID2 and ENY2 were mislocalized after loss of Tpr or GANP but not when Nup153, Nup50 or NXF1 were depleted (new Figure 4, and Supplementary Fig.7b,c). These data highlight a unique role of Tpr in anchoring TREX2 complex subunits to the NPC. CENT2-mCherry displayed broad nuclear localization and extremely dim signal at the nuclear envelope. Because we do not know whether mCherry-tagged CETN2 is functional, we decided to include only data from ENY2 and PCID2 in the manuscript.

Point 2a: Transcriptome changes are a rather poor functional readout because steady-state mRNA measurements are influenced by many factors. TPR loss appears to specifically impact RNAs, which are involved in regulation of transcription. This is potentially interesting, but what does it mean? The authors should follow-up on this observation.

Our data support the hypothesis that basket nucleoporin Tpr is an integral component of TREX2 machinery. It interacts with GANP, and required for GANP-mediated NPC localization of other TREX2 complex subunits. TREX2 subunits (ENY2 and GANP) interact with RNA polymerase II (PMID: 29329719) and binds to highly expressed genes, co-transcriptionally (PMID: 25294824). Moreover, GO analysis of Δ Sac3 (GANP) and Δ Thp (PCID2) subunit of yeast TREX2 complex showed enrichment of biological processes such as “transcription” and “mRNA processing” (PMID: 25294824), that were highlighted upon Tpr and GANP loss in our RNA-seq data, as well. We believe that GO enrichment in these terms can be explained by a role of TREX2 (and Tpr as its integral component) in coordination of transcription, 3'-end processing, and mRNA export processes that are tightly coupled together (PMID: 26709543). Therefore, loss of Tpr could have direct or indirect (through feedback loops) consequences on upstream steps of RNA processing and maturation, which is reflected in its GO-terms (see Discussion, p.13)

Point 2b: Similarly, I don’t see how the authors can conclude from comparing Ser5 ChIP-Seq data with mRNA levels that ffx1 is regulated at the post-transcriptional level. Overall, the authors should tone down their

conclusions about whether or not transcription rates are affected by loss of TPR / GANP. Methods for detecting nascent, actively transcribed Pol II transcripts are necessary to reach solid conclusion. This is beyond the scope of the paper, hence, a textual revision is appropriate.

We thank the reviewer for his/her insightful comment. Because this question was raised by both reviewers, we performed S⁴U anabolic labeling of newly synthesized RNAs with the following SLAM-seq that allowed us to detect nascent transcripts (new Figure 5 and 6). As the reviewer expected, Tpr loss indeed results in changes of transcriptional activities of Tpr-dependent genes. We made substantial revision of the text to incorporate these new data (Results p.10-11; Discussion p.12-13). Our new proposed model shows that Tpr is an integral component of TREX2 complex and is involved in both transcription and export of most of TREX2-dependent RNAs.

Point 3: What is the functional relationship between transcripts that are misregulated upon TPR/GANP depletion and those that are not properly exported in these cells? Are these distinct or overlapping sets of transcripts?

We thank the reviewer for this question. We analyzed RNA localization of the following genes: c-fos, gdf15, fjsx1 and hoxa13. After loss of either Tpr or GANP, c-fos and gdf15 were synthesized de novo but not effectively exported from the nucleus (Figure 6i; Supplementary Fig. 10g); whereas level of de novo synthesized fjsx1 and hoxa13 was reduced and remaining mRNA foci were resided within the nucleus (Figure 6k, Supplementary Fig. 10n). In addition, we performed RNA-FISH for several GANP- or GANP/NXF1-dependent but Tpr-independent genes (hist1h2ab, gadd45b, aen, and ubl4a) and found that all of them were retained in the nucleus after the loss of corresponding protein, as well (Supplementary Fig.7f, Supplementary Fig.11c, Results p.9, Discussion p.13). This, and the RNA-Seq data (Figure 2b) may suggest that TREX2(GANP) is a master regulator of both transcription and RNA maturation, while Tpr or NXF1 work as accessory proteins. Mis-processed RNA fail to exit the nucleus and potentially accumulate in nuclear speckles (Figure 3a).

Because we detect global accumulation of polyA RNAs inside nucleus after 8 h of Tpr loss, we suspect that nuclear retention in the absence of Tpr is a feature of many, if not all Tpr/GANP-dependent genes.

Interestingly, while our manuscript was under revision, a relevant paper from Dr. Palazzo group was deposited to bioRxiv (<https://doi.org/10.1101/740498>). By sequencing mRNA from the nuclear- and cytosol-enriched fractions of Tpr-depleted cells, they report that Tpr is required for the nuclear export of mRNAs that are generated from intronless and intron-poor genes. The relevant discussion was included into Discussion section (p.14).

Additionally, we also addressed the question of whether Tpr/GANP pathway overlaps with Crm1-dependent pathway, which is involved in export of small nuclear RNAs, ribosomal RNAs and a subset of mRNAs. We performed RNA-Seq of DLD-1 cells treated with Leptomycin B (100 nM, 3 h). We found a nonsignificant overlap between LMB-dependent genes and GANP-Tpr dependent genes (Supplementary Fig. 5h, Discussion p.12). Please see our new RNA-seq data deposition. To review GEO accession GSE148560:

1. <https://www.ncbi.nlm.nih.gov/geo/query/acc.cgi?acc=GSE148560>
2. Token: chmpyucejbjqfjef

Altogether, our results indicate that Tpr is a component of TREX2 complex and is involved in both transcription and export of many TREX2-dependent RNAs.

Point 4: Have the authors tried to ChIP-Seq TPR and GANP? This would be a major asset for the paper.

We prepared DNA samples from Tpr and GANP pulldowns; however, the amount and quality of DNA samples were below the expected quality of DNA samples required for ChIP-sequencing. Nonetheless, we proceeded with ChIP-sequencing of GANP samples but were not able to call peaks from the sequenced materials. Our discussions with Dr Marina Lusic (Heidelberg University Hospital) and several other groups during recent ASCB meeting indicated that ChIP-Seq of Tpr is a very challenging task. We agree that it is an important experiment but clearly different methodology (alternative protocols or antibodies) will need to be applied to answer this question.

Minor Point: The authors should better discuss why their data on the basket Nup targeting to the NPC core differs from earlier work. I still find the data confusing, particularly in comparison to the Umlauf et al., 2013 paper.

The appropriate discussion was included into Discussion section of the paper (please, see p.12-13).

The sequencing data of this study have been submitted to the NCBI Gene Expression Omnibus. Currently, the sequencing data are private.

- To review GSE132363 (RNA-seq) go to <https://www.ncbi.nlm.nih.gov/geo/query/acc.cgi?acc=GSE132363> (token: ivuxgssedbcdfuz).
- To review GSE148560 (RNA-seq, CRM1) use <https://www.ncbi.nlm.nih.gov/geo/query/acc.cgi?acc=GSE148560> (token: chmpyucejbqfjef).
- To review GSE149890 (SLAM-seq) use <https://www.ncbi.nlm.nih.gov/geo/query/acc.cgi?acc=GSE149890> (token: szqrawkuphmlch).
- To review BioProject ID: PRJNA548782 (ChIP Pol II Ser5P) go to <https://dataview.ncbi.nlm.nih.gov/object/PRJNA548782?reviewer=92gsntiq7mvua3u41jc1r97dbq>.

REVIEWERS' COMMENTS

Reviewer #1 (Remarks to the Author):

The revised manuscript by Dr Dasso and colleagues has very satisfactorily addressed my previous comments, specially regarding the mechanisms by which Tpr influences gene expression. I have only a few minor remarks that can be answered in the text:

* New Fig. 5 (SLAM-seq experiments).

- panel b: the authors should state in the figure legend that other conversion events (such as A>T, T>A, ...) are not detectable on their bar plot.
- panels d-e: could the authors define what is the Z-score ? I understand that this is a measurement of T>C conversion rates.

* I noted a few typos:

- Text p11, line 2: 1.4+-0.17 instead of 1.4+0.17
- Fig. 3f : GANP instead of GANP1

Reviewer #2 (Remarks to the Author):

The authors have addressed the majority of my concerns. The new data is technically sound and supports the original hypotheses.

I support its publication without further revisions.

Point-by-point response to the reviewers

Reviewers requests:

Reviewer #1:

The revised manuscript by Dr Dasso and colleagues has very satisfactorily addressed my previous comments, specially regarding the mechanisms by which Tpr influences gene expression. I have only a few minor remarks that can be answered in the text:

* New Fig. 5 (SLAM-seq experiments).

- panel b: the authors should state in the figure legend that other conversion events (such as A>T, T>A, ...) are not detectable on their bar plot.

- panels d-e: could the authors define what is the Z-score ? I understand that this is a measurement of T>C conversion rates.

* I noted a few typos:

- Text p11, line 2: 1.4+-0.17 instead of 1.4+0.17

- Fig. 3f : GANP instead of GANP1

Reviewer #2:

The authors have addressed the majority of my concerns. The new data is technically sound and supports the original hypotheses. I support its publication without further revisions.

Authors' response:

Thank you for the detailed review of our manuscript.

We added the information about other than T>C conversion events to figure legend of Figure 5. P.30-31.

We added the information how Z-score have been defined to the figure legend of Figure 5. In addition, we added information for how the Z-score was quantified into Materials and Method section, P.20-21.

Text on P.11 was corrected.

We corrected the Figure 3f.

Thank you for the detailed review of our manuscript.